# Impact of Computation in Integral Reinforcement Learning for Continuous-Time Control

**Wenhan Cao**[1,2]  **Wei Pan**[1]
[1]Department of Computer Science, University of Manchester
[2]School of Vehicle and Mobility, Tsinghua University
`cwh19@mails.tsinghua.edu.cn`  `wei.pan@manchester.ac.uk`

## Abstract

Integral reinforcement learning (IntRL) demands the precise computation of the utility function's integral at its policy evaluation (PEV) stage. This is achieved through quadrature rules, which are weighted sums of utility functions evaluated from state samples obtained in discrete time. Our research reveals a critical yet underexplored phenomenon: the choice of the computational method – in this case, the quadrature rule – can significantly impact control performance. This impact is traced back to the fact that computational errors introduced in the PEV stage can affect the policy iteration's convergence behavior, which in turn affects the learned controller. To elucidate how computation impacts control, we draw a parallel between IntRL's policy iteration and Newton's method applied to the Hamilton-Jacobi-Bellman equation. In this light, computational error in PEV manifests as an extra error term in each iteration of Newton's method, with its upper bound proportional to the computational error. Further, we demonstrate that when the utility function resides in a reproducing kernel Hilbert space (RKHS), the optimal quadrature is achievable by employing Bayesian quadrature with the RKHS-inducing kernel function. We prove that the local convergence rates for IntRL using the trapezoidal rule and Bayesian quadrature with a Matérn kernel to be $O(N^{-2})$ and $O(N^{-b})$, where $N$ is the number of evenly-spaced samples and $b$ is the Matérn kernel's smoothness parameter. These theoretical findings are finally validated by two canonical control tasks.

## 1 Introduction

Recent advancements in reinforcement learning (RL) have prominently focused on discrete-time (DT) systems. Notable applications range from Atari games Schrittwieser et al. (2020) to the game of Go Silver et al. (2016; 2017) as well as large language models Bubeck et al. (2023). However, most physical and biological systems are inherently continuous in time and driven by differential equation dynamics. This inherent discrepancy underscores the need for the evolution of continuous-time RL (CTRL) algorithms Baird (1994); Lewis et al. (1998); Abu-Khalaf & Lewis (2005); Vrabie & Lewis (2009); Vrabie et al. (2009); Lewis & Vrabie (2009); Vamvoudakis & Lewis (2010); Modares et al. (2014); Lee et al. (2014); Modares & Lewis (2014); Vamvoudakis et al. (2014); Yildiz et al. (2021); Holt et al. (2023); Wallace & Si (2023).

Unfortunately, adopting CTRL presents both conceptual and algorithmic challenges. First, Q-functions are known to vanish in CT systems Baird (1994); Abu-Khalaf & Lewis (2005); Lewis & Vrabie (2009), which makes even simple RL algorithms such as Q-learning infeasible for CT systems. Second, the one-step transition model in DT system needs to be replaced with time derivatives, which leads to the CT Bellman equation (also called nonlinear Lyapunov equation or generalized Hamilton–Jacobi Bellman equation) governed by a complex partial differential equation (PDE) rather than a simple algebraic equation (AE) as in DT systems Lewis & Vrabie (2009); Vrabie & Lewis (2009); Vamvoudakis & Lewis (2010):

$$\textbf{CT Bellman Equation (PDE):} \qquad \dot{V}^u(x(t)) = -l(x(t), u(x(t))), \qquad (1a)$$

$$\textbf{DT Bellman Equation (AE):} \qquad \Delta V^u(x(k)) = -l(x(k), u(x(k))). \qquad (1b)$$

In (1a) and (1b), the symbols $x, u, l, V$ represent the state, control policy, utility function and value function, respectively. While traditional DTRL focuses on maximizing a "reward function", our

work, in line with most CTRL literature, employs a "utility function" to represent costs or penalties, aiming to minimize its associated value. Solving the Bellman equation refers to the policy evaluation (PEV), which is a vital step in policy iteration (PI) of RL Sutton et al. (1998); Lewis & Vrabie (2009). However, CT Bellman equation cannot be solved directly because the explicit form of $l(x(t), u(x(t)))$ hinges on the explicit form of the state trajectory $x(t)$, which is generally unknown.

The CT Bellman equation can be formulated as the interval reinforcement form Vrabie & Lewis (2009); Lewis & Vrabie (2009); Modares et al. (2014); Modares & Lewis (2014); Vamvoudakis et al. (2014), which computes the value function $V$ through the integration of the utility function:

$$V^u(x(t)) = \xi(l) + V^u(x(t + \Delta T)), \quad \xi(l) = \int_t^{t+\Delta T} l(x(s), u(s)) \, \mathrm{d}s. \tag{2}$$

Here, $\xi(l)$ represents the integral of the utility function. When the system's internal dynamics is unknown, $\xi(l)$ can only be computed by a quadrature rule solely utilizing state samples in discrete time, i.e.,

$$\xi(l) \approx \hat{\xi}(l) = \sum_{i=1}^{N} w_i l(x(t_i), u(x(t_i))). \tag{3}$$

Here, the time instants for collecting state samples are $t_1 = t < t_2 < \cdots < t_N = t + \Delta T$, with $N$ being the sample size. The quadrature rule is characterized by a set of weights $\{w_i\}_{i=1}^N$, which can be chosen from classical methods like the trapezoidal rule or advanced probabilistic methods such as Bayesian quadrature (BQ) O'Hagan (1991); Karvonen & Särkkä (2017); Cockayne et al. (2019); Briol et al. (2019); Hennig et al. (2022). For simplicity, we denote the computational error for the quadrature rule as $\mathrm{Err}(\hat{\xi}(l)) := |\xi(l) - \hat{\xi}(l)|$. In real-world applications, sensors of autonomous systems are the primary source of state samples. For example, state samples are gathered from various sensors when the CTRL algorithm trains a drone navigation controller. These sensors include the gyroscope for orientation, the accelerometer for motion detection, the optical flow sensor for positional awareness, the barometer for altitude measurement, and GPS for global positioning. If these sensors operate at a sampling frequency of 10 Hz and the time interval $\Delta T$ in (2) is set to 1 second, we would obtain $N = 11$ state samples within that duration.

The impact of computational methods on the solution of the CT Bellman equation is emphasized by Yildiz et al. (2021). Echoing this finding, our study further reveals that the choice of computational method can also impact the performance of the learned controller. As illustrated in Figure 1, we investigate the performance of the Integral Reinforcement Learning (IntRL) algorithm Vrabie & Lewis (2009). We utilize both the trapezoidal rule and the BQ with a Matérn kernel to compute the PEV step, applying different sample sizes $N$. Our results for a canonical control task (Example 1 of Vrabie & Lewis (2009)) indicate that larger sample sizes diminish accumulated costs. In addition, we observe notable differences in accumulated costs between these two quadrature methods. This trend highlights a crucial insight: **the computational method itself can be a determining factor in the control performance**. This phenomenon is not exclusive to the IntRL algorithm but applies to the CTRL algorithm when internal dynamics is known, as elaborated in Appendix A.

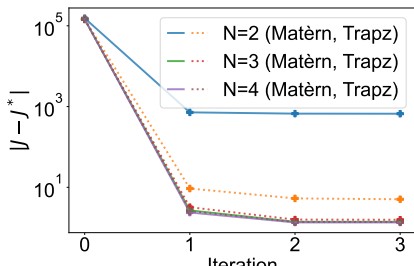

Figure 1: Evaluation of accumulated cost $J$ for controllers, computed through the trapezoidal rule and BQ with a Matérn kernel, compared to the optimal controller cost $J^*$. This simulation is performed on different sample sizes $N$ and relies on evenly-spaced samples within a canonical control task in Vrabie & Lewis (2009).

The impact of computational methods — specifically, the quadrature rule as described in (3) for unknown internal dynamics scenarios — on control performance is a compelling yet underexplored topic. This gap prompts crucial questions: How does the computation impact control performance? How can we quantitatively describe this impact? Despite the importance of these questions, existing literature has yet to address them. Our paper aims to bridge this gap by investigating the intricate relationship between computational methods and controller performance. We choose to focus our analysis on the IntRL algorithm Vrabie & Lewis (2009). This choice is motivated by IntRL's role as a foundational CTRL algorithm in unknown internal dynamics scenarios Modares et al. (2014); Lee et al. (2014); Modares & Lewis (2014); Vamvoudakis et al. (2014); Wallace & Si (2023). As

we will illustrate in Section 2, understanding the impact of computation on control in scenarios with unknown internal dynamics is more crucial and merits deeper examination.

The contributions of this paper can be summarized as follows: **(1)** We show that the PEV step in IntRL essentially computes the integral of the utility function. This computation relies solely on state samples obtained at discrete time intervals, introducing an inherent computational error. This error is bounded by the product of the integrand's norm in the reproducing kernel Hilbert space (RKHS), and the worst-case error. When employed as the quadrature rule, BQ minimizes the worst-case error, which coincides precisely with BQ's posterior covariance. Additionally, we analyze the computational error's convergence rate concerning the sample size, focusing on both Wiener and Matérn kernels for evenly spaced samples. **(2)** We demonstrate that PI of IntRL can be interpreted as Newton's method applied to the Hamilton-Jacobi-Bellman (HJB) equation. Viewed from this perspective, the computational error in PEV acts as an extra error term in each Newton's method iteration, with its upper bound proportional to the computational error. **(3)** We present the local convergence rate of PI for IntRL, in terms of sample size, for both the trapezoidal rule and BQ using the Matérn kernel. Specifically, the convergence rate for PI is $O(N^{-2})$ when using the trapezoidal rule, and $O(N^{-b})$ when using BQ with a Matérn kernel, where $b$ is the smoothness parameter. Our paper is the first to explore the impact of computational errors on the controller learned by IntRL.

The remainder of this paper is organized as follows: Section II provides a detailed problem formulation. Our main results are presented in Section III, followed by simulations in Section IV. Section V draws conclusions and discussions on this new topic.

## 2 PROBLEM FORMULATION

We assume that the governing system is control affine and the system's internal dynamics $f$ is not necessarily known throughout the paper:

$$\dot{x} = f(x) + g(x)u, \tag{4}$$

where $x \in \Omega \subseteq \mathbb{R}^{n_x}$ is the system state, $u \in \mathbb{R}^{n_u}$ is the control input, $f : \mathbb{R}^{n_x} \to \mathbb{R}^{n_x}$ and $g : \mathbb{R}^{n_x} \to \mathbb{R}^{n_x \times n_u}$ are locally Lispchitz mappings with $f(0) = 0$. The objective is to find a control policy $u$ that minimizes the performance index:

$$J(x_0, u) = \int_0^\infty l(x(s), u(s)) \, \mathrm{d}s, \quad x(0) = x_0, \tag{5}$$

where $l(x, u) = q(x) + u^\top R(x) u$ is the utility function, with $q(x)$ and $R(x)$ being the positive definite function and positive definite matrix, respectively. Furthermore, the system (4) is assumed to be zero-state observable through $q$ Beard et al. (1997). The optimal control policy $u^*$ and its associated optimal value function $V^*$ are defined as:

$$V^*(x_0) := \min_{u \in A(\Omega)} J(x_0, u), \quad u^*(x_0) := \arg\min_{u \in A(\Omega)} J(x_0, u). \tag{6}$$

Here, $u \in A(\Omega)$ represents that $u$ is an admissible policy on $\Omega$. For more details of the definition of an admissible policy, please refer to Appendix B. The optimal value function can be determined by the HJB equation, which is illustrated in the following theorem:

**Theorem 1** (HJB Equation Properties Vrabie & Lewis (2009))**.** *A unique, positive definite, and continuous function $V^*$ serves as the solution to the HJB equation:*

**HJB Equation:** $\quad G(V^*) = 0, \quad G(V) := (\nabla_x V)^\top f - \frac{1}{4} (\nabla_x V)^\top g R^{-1} g^\top \nabla_x V + q. \tag{7}$

*In this case, $V^*$ is the optimal value function defined in (6). Consequently, the optimal control policy can be represented as $u^*(x) = -\frac{1}{2} R^{-1} g(x)^\top \nabla_x V^*$.*

While solving the HJB equation poses a challenge as it lacks an analytical solution, IntRL tackles this issue by utilizing PI to iteratively solve the infinite horizon optimal control problem. This approach is taken without necessitating any knowledge of the system's internal dynamics $f$ Vrabie & Lewis (2009). Consider $u^{(0)}(x(t)) \in A(\Omega)$ be an admissible policy, PI of IntRL performs PEV and policy improvement (PIM) iteratively:

**PEV:** $\quad V^{(i)}(x(t)) = \int_t^{t+\Delta T} l(x(s), u^{(i)}(x(s))) \, \mathrm{d}s + V^{(i)}(x(t + \Delta T)), \; V^{(i)}(x) = 0, \tag{8a}$

**PIM:** $\quad u^{(i+1)}(x) = -\frac{1}{2} R^{-1} g(x)^\top \nabla_x V^{(i)}. \tag{8b}$

In the PI process, the PEV step (8a) assesses the performance of the currently derived control policy, while the PIM step (8b) refines this policy based on the evaluated value function. According to Vrabie & Lewis (2009), these iterative steps converge to an optimal control policy and its corresponding optimal value function when computations, i.e., the integration of the utility function $\xi(l)$, are performed flawlessly. However, any computational error $\text{Err}(\hat{\xi}(l))$ of the quadrature rule in (3) can adversely affect the accuracy of PEV. This inaccuracy can subsequently influence the updated control policies during the PIM step. Such computational errors may accumulate over successive iterations of the PI process and finally impact the learned controller, leading to the phenomenon of "computation impacts control" in IntRL.

The impact of computation requires more consideration for systems with unknown internal dynamics. When the system's internal dynamics is known, adaptive numerical solvers such as the Runge-Kutta series, offer an accurate solution to the CT Bellman equation (refer to Appendix C). These solvers adjust their step sizes according to the local behavior of the equation they solve, promising high computational accuracy. However, this step-size adjustment mechanism becomes a limitation in unknown internal dynamics cases. For instance, the Runge-Kutta (4,5) method produces both fourth- and fifth-order approximations, and the difference between the two approximations refines the step size. In real-world autonomous systems, state samples are typically acquired from sensors at evenly spaced intervals, challenging alignment with adaptive solver steps. Hence, adaptive numerical solvers are not suitable for unknown internal dynamics. In such cases, approximation methods such as quadrature rules become essential, but might introduce significant computational errors in the PEV step. This fact underscores the importance of considering the computational impact on control performance when facing unknown internal dynamics.

Given the preceding discussion, it is evident that computational errors significantly influence control performance by affecting the convergence behavior of PI. With this understanding as the backdrop, this paper aims to examine the convergence rate of PI for IntRL in the presence of computational errors. Specifically, we seek to address two key questions: **(1)** How does the computational error $\text{Err}(\hat{\xi}(l))$ in the PEV step of PI impact the IntRL algorithm's convergence? **(2)** How to quantify the computational error $\text{Err}(\hat{\xi}(l))$ in the PEV step and what is the optimal quadrature rule?

## 3 THEORETICAL ANALYSIS

### 3.1 CONVERGENCE ANALYSIS OF PI

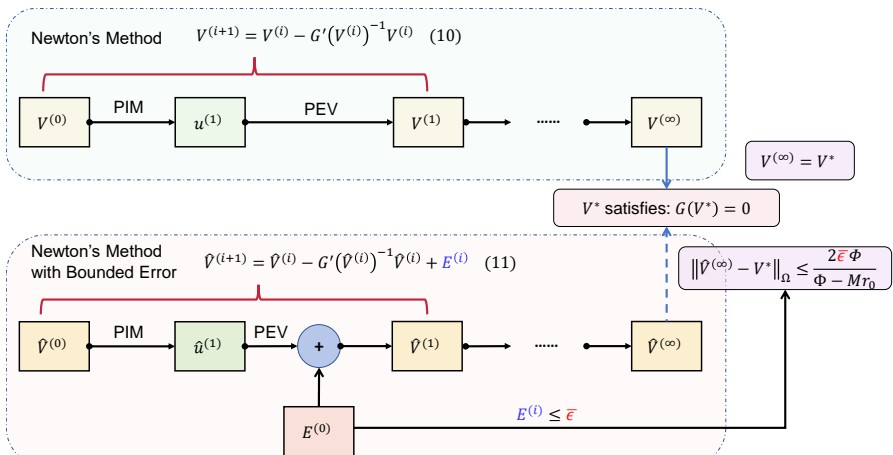

Figure 2: Relationship between PI and Newton's method. The standard PI can be regarded as performing Newton's method to solve the HJB equation, while PI incorporated by the computational error can be seen as Newton's method with bounded error.

In this subsection, we answer the first question, i.e., we examine the impact of computational error in the PEV step on the convergence behavior of IntRL. Initially, we utilize the Fréchet differential to demonstrate the correspondence between the standard PI process in IntRL (without computational error) and Newton's method within the Banach space to which the value function belongs.

**Lemma 1** (PI as Newton's Method in Banach Space). *Consider a Banach space* $\mathbb{V} \subset \{V(x)|V(x) : \Omega \to \mathbb{R}, V(0) = 0\}$, *equipped with the norm* $\| \cdot \|_\Omega$. *For* $G(V)$ *as defined in* (7), *its Gâteaux and Fréchet derivatives at* $V$ *can be expressed as*

$$G'(V)W = \frac{\partial W}{\partial x^\top} f - \frac{1}{2} \frac{\partial W}{\partial x^\top} g R^{-1} g^\top \frac{\partial V}{\partial x}. \tag{9}$$

*Consider the value function* $V^{(i)}$ *at the* $i^{th}$ *iteration of PI, executing PIM* (8b) *and PEV* (8a) *iteratively corresponds to Newton's method applied to solve the HJB equation* (7). *This Newton's method iteration is formulated as:*

$$V^{(i+1)} = V^{(i)} - \left[ G'(V^{(i)}) \right]^{-1} G(V^{(i)}). \tag{10}$$

The proof of the lemma is presented in Appendix D. In this lemma, we demonstrate that a single iteration consisting of both the PIM and PEV in IntRL is equivalent to one iteration of Newton's method for solving the HJB equation in a Banach space. This parallel allows us to view the computational error introduced in the PEV step of the $i^{\text{th}}$ iteration as an extra error term, denoted $E^{(i)}$, in Newton's method iteration. We define the value function with the incorporated error term $E^{(i)}$ (highlighted in blue) as $\hat{V}^{(i)}$. Its iterative update is given by:

$$\hat{V}^{(i+1)} = \hat{V}^{(i)} - \left[ G'(\hat{V}^{(i)}) \right]^{-1} G(\hat{V}^{(i)}) + E^{(i)}, \quad \hat{V}^{(0)} = V^{(0)}. \tag{11}$$

Assume that the error term $E^{(i)}$ has a bounded norm, denoted as $\|E^{(i)}\|_\Omega \le \bar{\epsilon}$, where the bound $\bar{\epsilon}$ is proportional to the computational error introduced during the PEV step, which will be shown later in Section 3.3. The subsequent theorem describes the convergence behavior of Newton's method when an extra bounded error term is included.

**Theorem 2** (Convergence of Newton's Method with Bounded Error). *Define* $B_0 = \{V \in \mathbb{V} : \|V - V^{(0)}\|_\Omega \le r_0\}$, $B = \{V \in \mathbb{V} : \|V - V^{(0)}\|_\Omega \le r_0 + d\}$ *with* $d \ge 0$, $r_0 = \| \left[ G'(V^{(0)}) \right]^{-1} G(V^{(0)}) \|_\Omega$, *and* $L_0 = \sup_{V \in B_0} \| \left[ G'(V^{(0)}) \right]^{-1} G''(V) \|_\Omega$. *If we assume the following conditions: (i)* $G$ *is twice Fréchet continuous differentiable on* $B$; *(ii)* $r_0 L_0 \le \frac{1}{2}$; *(iii)* $G'(V)$ *is nonsingular for* $\forall V \in B$; *(iv) There exists a number* $\Phi, M > 0$ *such that* $\frac{1}{\Phi} \ge \sup_{V \in B} \| [G'(V)]^{-1} \|_\Omega$, $M \ge \sup_{V \in B} \|G''(V)\|_\Omega$ *and* $\Phi - M r_0 > \sqrt{2 M \bar{\epsilon} \Phi}$; *(v)* $d > \frac{2 \bar{\epsilon} \Phi}{\Phi - M r_0}$. *Then we have* $\hat{V}^{(i)} \in B$ *and*

$$\|\hat{V}^{(i)} - V^*\|_\Omega \le \frac{2 \bar{\epsilon} \Phi}{\Phi - M r_0} + \frac{2^{-i} (2 r_0 L_0)^{2^i}}{L_0}, \tag{12}$$

*where* $V^*$ *is optimal value function.*

The proof can be found in Appendix E. Theorem 2 clarifies that the convergence behavior of Newton's method, when augmented with an extra bounded error term, hinges on two key elements: the cumulative effect of the bounded error term over the iterations and the inherent convergence properties of the standard PI algorithm. It should be noted that the convergence property described in Theorem 2 is local and applies when the underlying operator $G(V)$ is twice continuously differentiable. This iteration commences with an estimate of the constant $d$ to ascertain the value of $B$. If condition (iv) of the theorem is not met, we are likely encountering a pathological situation, which necessitates choosing a smaller $d$ until the condition (iv) is satisfied or (iv) and (v) are found to be incompatible in which case our analysis breaks down. This theorem extends existing insights from earlier work on the error analysis of Newton's method Urabe (1956), which assumes a strict Lipschitz condition for the operator $G$. However, this Lipschitz condition can be difficult to verify for utility functions $l$ that do not include a time discount factor. A comprehensive representation of the convergence analysis conducted in this subsection is provided in Figure 2.

## 3.2 COMPUTATIONAL ERROR QUANTIFICATION

After analyzing the convergence property, our subsequent focus is quantifying the computational error inherent to each PEV step of IntRL. As discussed in Section 1, when internal dynamics is unknown, the integral of the utility function $\xi(l)$ (2) in the PEV step must be approximated using state samples in discrete time. This approximation will inevitably introduce computational integration error, especially when dealing with sparse samples, such as those collected from sensors on

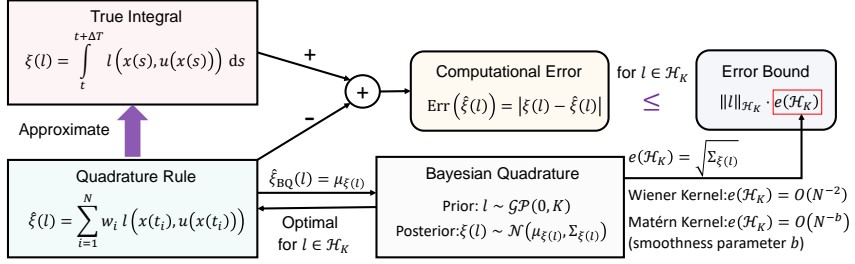

Figure 3: Flowchart illustrating the quantification of computational error. The computational error is defined as the absolute difference between the true integral and its approximation derived from the quadrature rule. The computational error is bounded by the product of the integrand's norm in the RKHS, and the worst-case error. When employed as the quadrature rule, BQ minimizes the worst-case error, which coincides precisely with BQ's posterior covariance.

real-world autonomous systems. A tight upper bound of the computational error, given a specific quadrature rule $\hat{\xi}(l)$, can be determined by the Cauchy-Schwarz inequality, provided that $l$ belongs to the RKHS $\mathcal{H}_K$ Kanagawa et al. (2016; 2020); Briol et al. (2019):

$$\text{Err}\left(\hat{\xi}(l)\right) \leq \|l\|_{\mathcal{H}_K} \cdot e(\mathcal{H}_\mathcal{K}), \quad e(\mathcal{H}_\mathcal{K}) := \sup_{l \in \mathcal{H}_\mathcal{K}, \|l\|_{\mathcal{H}_K} \leq 1} \text{Err}\left(\hat{\xi}(l)\right). \tag{13}$$

Here, $\mathcal{H}_K$ represents the RKHS induced by the kernel function $K$ with $\|\cdot\|_{\mathcal{H}_K}$ being its corresponding norm, and $e(\mathcal{H}_\mathcal{K})$ is the worst-case error. From (13), it is clear that if a quadrature rule satisfies $e(\mathcal{H}_\mathcal{K}) = O(N^{-b})$ with $b > 0$, then the computational error satisfies $\text{Err}\left(\hat{\xi}(l)\right) = O(N^{-b})$. Next, we will discuss which quadrature rule minimizes the worst-case error.

**Minimum of the worst case error – BQ:** The optimal quadrature rule that minimizes the worst-case error $e(\mathcal{H}_K)$ is the maximum a posteriori (MAP) estimate of BQ Briol et al. (2019). Specifically, the minimization can be achieved when the kernel function of BQ is set as the kernel function $K$ that induces RKHS $\mathcal{H}_K$. BQ aims to approximate the integral $\xi(l)$ by specifying a Gaussian process prior to the integrand (in this case, the integrand is the utility function $l$) denoted as $l \sim \mathcal{GP}(0, K)$ and conditioning this prior on the integrand evaluated at the discrete time $\{l(x(t_i), u(x(t_i)))\}_{i=1}^{N}$ to obtain the Gaussian posterior $\xi(l) \sim \mathcal{N}\left(\mu_{\xi(l)}, \Sigma_{\xi(l)}\right)$ O'Hagan (1991); Karvonen & Särkkä (2017); Cockayne et al. (2019); Briol et al. (2019); Hennig et al. (2022):

$$\mu_{\xi(l)} = \int_t^{t+\Delta T} K(s, T)^{-1}\, \mathrm{d}s\, K_{TT}^{-1}\, l(T),$$

$$\Sigma_{\xi(l)} = \int_t^{t+\Delta T} \int_t^{t+\Delta T} \left[K(s, s') - K(s, T)K_{TT}^{-1}K(T, s')\right]\, \mathrm{d}s\, \mathrm{d}s'.$$

Here, $T = \{t_i\}_{i=1}^{N}$ represents the set of time instants, and $l(T)$ corresponds to the evaluated values of the utility function at these sample points, with $(l(T))_i = l(x(t_i), u(x(t_i)))$ being the evaluation at the $i^{\text{th}}$ sample point. In addition, $(K(\cdot, T))_i := K(\cdot, t_i)$ and $(K_{TT})_{ij} := K(t_i, t_j)$. Note that the posterior mean $\mu_{\xi(l)}$ takes the form of a quadrature rule, with weights $w_i$ being the elements of the vector $\int_t^{t+\Delta T} K(s, T)^{-1}\, \mathrm{d}s\, K_{TT}^{-1}$. The BQ estimate of $\xi(l)$ can be obtained using the MAP estimate, i.e., the posterior mean $\hat{\xi}_{\text{BQ}}(l) = \mu_{\xi(l)}$. In this case, the worst-case error equals the square root of the posterior variance $e(\mathcal{H}_K) = \sqrt{\Sigma_{\xi(l)}}$ Kanagawa et al. (2016; 2020); Briol et al. (2019).

**BQ with Wiener Kernel (Trapezoidal Rule)** When the kernel function of the BQ is chosen as the Wiener process (we call as Wiener kernel for simplicity), i.e., $K_{\text{Wiener}}(s, s') := \max\{s, s'\} - s''$ with $s'' < t$, the posterior mean and covariance of BQ are calculated as Hennig et al. (2022):

$$\mu_{\xi(l)} = \sum_{i=1}^{N-1} \frac{l(x(t_i), u(x(t_i))) + l(x(t_{i+1}), u(x(t_{i+1})))}{2}\delta_i, \quad \Sigma_{\xi(l)} = \frac{1}{12}\sum_{i=1}^{N-1} \delta_i^3.$$

Here, $\delta_i := t_{i+1} - t_i$. The BQ estimate is equal to the integral approximated by the trapezoidal rule Sul'din (1959); Hennig et al. (2022). For data points evenly spaced at the interval, i.e., $\delta_i = \frac{\Delta T}{N-1}$, the

posterior variance of BQ becomes $\Sigma_{\xi(l)} = \frac{\Delta T^3}{12(N-1)^2}$, implying a linear convergence rate $O(N^{-1})$ of the computational error bound when $l$ belongs to the RKHS induced by Wiener kernel. However, this might not represent a tight upper bound, since the convergence rate of the trapezoidal rule can achieve $O(N^{-2})$ when the integrand function is twice differentiable Atkinson (1991).

**BQ with Matérn kernel (In Sobolev Space)** The Sobolev space $W_2^b$ of order $b \in \mathbb{N}$ is defined by $W_2^b := \{\mathcal{F} \in L_2 : D^\alpha \mathcal{F} \in L_2 \text{ exists } \forall |\alpha| \leq b\}$, where $\alpha := (\alpha_1, \alpha_2, ..., \alpha_d)$ with $\alpha_k \geq 0$ is a multi-index and $|\alpha| = \sum_{k=1}^d \alpha_k$ Adams & Fournier (2003). Besides, $D^\alpha \mathcal{F}$ is the $\alpha^{\text{th}}$ weak derivative of $\mathcal{F}$ and its norm is defined by $\|\mathcal{F}\|_{W_2^b} = \left(\sum_{|\alpha| \leq b} \|D^\alpha \mathcal{F}\|_{L_2}^2\right)^{\frac{1}{2}}$. The Sobolev space $W_2^b$ is the RKHS with the reproducing kernel $K$ being the Matérn kernel with smoothness parameter $b$ Seeger (2004); Matérn (2013); Kanagawa et al. (2020); Pagliana et al. (2020). More details about the Matérn kernel can be found in Appendix F. In addition, the computational error for the optimal quadrature rule, i.e., BQ with Matérn kernel, achieves $O(N^{-b})$ if $l \in W_2^b$ Novak (2006); Kanagawa et al. (2020). The illustration of BQ can be found in Appendix G and the technical flowchart of computational error quantification can be summarized in Figure 3.

## 3.3 CONVERGENCE RATE OF INTRL FOR DIFFERENT QUADRATURE RULES

After illustrating how to quantify the computational error, we will delve deeper into understanding its implications on the solution of PEV (8a). And, more importantly, we aim to uncover its impact on the convergence of PI. Consider the case where the value function is approximated by the linear combination of basis functions Vrabie & Lewis (2009); Vamvoudakis et al. (2014):

$$V^{(i)}(x) = \omega^{(i)\top} \phi(x), \quad \phi(x) := [\phi_1(x), \phi_2(x), ..., \phi_{n_\phi}(x)]^\top. \tag{14}$$

Here, $\phi(x)$ represent the basis functions with $\phi_k(0) = 0, \forall k = 1, 2, ..., n_\phi$, and $\omega^{(i)} \in \mathbb{R}^{n_\phi}$ denotes the parameters determined at the $i^{\text{th}}$ iteration. A typical assumption of value function and the basis function is expressed as Vrabie & Lewis (2009); Vamvoudakis et al. (2014):

**Assumption 1** (Continuity, Differentiability and Linear Independence). *The value function obtained at each PEV iteration (8a), is continuous and differentiable across $\Omega$, denoted by $V^{(i)} \in \mathbb{C}^1(\Omega)$. Besides, $\phi_k(x) \in \mathbb{C}^1(\Omega)$, $\forall k = 1, 2, ..., n_\phi$, and the set $\{\phi_k(x)\}_1^{n_\phi}$ are linearly independent.*

Such an assumption is commonly encountered in optimal control Lewis et al. (1998); Abu-Khalaf & Lewis (2005); Vrabie & Lewis (2009); Vrabie et al. (2009), suggesting that the value function can be uniformly approximated by $\phi$ when its dimension $n_\phi$ approaches infinity. In our analysis, we focus primarily on the impact of computational error. To isolate this effect, we make the assumption that the learning errors introduced by the approximation in (14) can be neglected. This assumption is based on the premise that such errors can be effectively minimized or rendered negligible through adequate training duration and optimal hyperparameter tuning, among other techniques. With this approximation (14), the parameter $\omega^{(i)}$ is found via the linear matrix equation:

$$\Theta^{(i)}\omega^{(i)} = \Xi^{(i)}, \quad \Xi^{(i)} := \begin{bmatrix} \xi(T_1, T_2, u^{(i)}) \\ \xi(T_2, T_3, u^{(i)}) \\ ... \\ \xi(T_m, T_{m+1}, u^{(i)}) \end{bmatrix}, \quad \Theta^{(i)} := \begin{bmatrix} \phi^\top(x(T_2)) - \phi^\top(x(T_1)) \\ \phi^\top(x(T_3)) - \phi^\top(x(T_2)) \\ ... \\ \phi^\top(x(T_{m+1})) - \phi^\top(x(T_m)) \end{bmatrix}.$$
$$\tag{15}$$

Here, $\Theta^{(i)}$ has full column rank given approximate selections of $T_1, T_2, ..., T_{m+1}$. This condition can be seen as the persistence condition Wallace & Si (2023), ensuring that equation (15) yields a unique solution. For more discussions, see Appendix H. Besides, $\xi(T_k, T_{k+1}, u^{(i)}) := \int_{T_k}^{T_{k+1}} l(x(s), u^{(i)}(x(s))) \, \mathrm{d}s$ represents the integral over the time interval from $T_k$ to $T_{k+1}$ of the utility function $l$. As previously discussed, achieving the exact value of this integral is not feasible. It is instead approximated using a quadrature rule, with its computational error bounded by (13). We symbolize the computational error's upper bound for $\xi(T_k, T_{k+1}, u^{(i)})$ as $\delta\xi(T_k, T_{k+1}, u^{(i)})$. Therefore, the integral value for $\xi(T_k, T_{k+1}, u^{(i)})$ lies within the interval $[\hat{\xi}(T_k, T_{k+1}, u^{(i)}) - \delta\xi(T_k, T_{k+1}, u^{(i)}), \hat{\xi}(T_k, T_{k+1}, u^{(i)}) + \delta\xi(T_k, T_{k+1}, u^{(i)})]$. Here, $\hat{\xi}(T_k, T_{k+1}, u^{(i)})$ signifies the integral's estimate using the quadrature rule. For simplicity, this can be formulated in matrix form as: $\Xi^{(i)} \in [\hat{\Xi}^{(i)} - \delta\Xi^{(i)}, \hat{\Xi}^{(i)} + \delta\Xi^{(i)}]$. Consequently, due to the computational error, the actual linear

matrix equation we solve in the PEV step becomes (16) instead of (15):

$$\hat{\Theta}^{(i)}\hat{\omega}^{(i)} = \hat{\Xi}^{(i)}. \tag{16}$$

Here, $\hat{\Theta}^{(i)}$ is the data matrix constructed by the states generated by the control policy $\hat{u}^{(i)} := -\frac{1}{2}R^{-1}g(x)^{\top}\nabla_x \hat{V}^{(i-1)}$. Incorporating the convergence analysis of PI in Section 3.1 with the computational error quantification for each PI step in Section 3.2, we can present our main results.

**Theorem 3** (Convergence Rate of IntRL Concerning Computational Error). *Assume that $\Omega$ is a bounded set, and the conditions outlined in Theorem 2 hold. Under Assumption 1, we have:*

$$|\hat{V}^{(i)}(x) - V^*(x)| \le \frac{2\Phi\bar{\epsilon}}{\Phi - Mr_0} + \frac{2^{-i}(2r_0L_0)^{2^i}}{L_0}, \quad \forall x \in \Omega. \tag{17}$$

*Here, $\bar{\epsilon} = \|\phi\|_{\infty} \cdot \sup_i \left\{ \|(\hat{\Theta}^{(i)\top}\hat{\Theta}^{(i)})^{-1}\hat{\Theta}^{(i)\top}\|_2 \|\delta\Xi^{(i)}\|_2 \right\}$ is the upper bound of the extra error term $E^{(i)}$ in (11), satisfying $E^{(i)} \le \bar{\epsilon}$, where $\|\phi\|_{\infty} := \max_{1 \le k \le n_\phi} \sup_{x \in \Omega} |\phi_k(x)|$. Besides, the definitions of $\Phi, M, r_0, L_0$ are consistent with Theorem 2.*

The proof is given in appendix I. In (17), $\delta\Xi^{(i)}$ is a vector with its $k^{\text{th}}$ row defined as $\delta\xi(T_k, T_{k+1}, \hat{u}^{(i)})$, serving as an upper bound for the computational error of the integral $\xi(T_k, T_{k+1}, \hat{u}^{(i)}) = \int_{T_k}^{T_{k+1}} l(x(s), \hat{u}^{(i)}(x(s)))\mathrm{d}s$. By invoking Theorem 3, we can relate the computational error $\delta\xi(T_k, T_{k+1}, \hat{u}^{(i)})$ to the upper bound of the extra error term $\bar{\epsilon}$ in the modified Newton's method (11). This relationship establishes the convergence rate of IntRL. Combining with prior discussions of the computational error for the trapezoidal rule and the BQ with Matérn kernel, we can end our analysis with the subsequent corollary:

**Corollary 1** (Convergence Rate of IntRL for Trapezoidal Rule and BQ with Matern Kernel). *Assuming that the sample points are uniformly distributed within each time interval $[T_k, T_{k+1}]$ for $k = 1, 2, ..., m$, and that the IntRL algorithm has run for sufficient iterations (i.e., $i \to \infty$). Under the conditions in Theorem 3, if $l$ belongs to the RKHS induced by the Wiener kernel and the integrals in the PEV step are calculated by the trapezoidal rule, we have $|\hat{V}^{(\infty)}(x) - V^*(x)| = O(N^{-2})$. Here, $\hat{V}^{(\infty)}(x)$ is defined as $\hat{V}^{(\infty)}(x) := \lim_{i \to \infty} \hat{V}^{(i)}(x)$. If $l$ belongs to the RKHS $W_2^b$ induced by the Matérn kernel and the integrals in the PEV step are calculated by the BQ with Matérn kernel (having smoothness parameter b), we have $|\hat{V}^{(\infty)}(x) - V^*(x)| = O(N^{-b})$.*

## 4 EXPERIMENTAL RESULTS

In this section [1], we validate the convergence rate of the IntRL for the trapzoidal rule and BQ with Matérn kernel as demonstrated in Corollary 1.

**Example 1 (Linear System):** First, we will consider the canonical linear-quadratic regulator problem defined by system dynamics $\dot{x} = Ax + Bu$ and the utility function $l(x, u) = x^{\top}Qx + u^{\top}Ru$. The associated PI can be obtained by setting $\phi(x) = x \otimes x$. The parameter of value function $\omega^{(i)}$ equals $\mathrm{vec}(P_i)$, where $P_i$ is the solution to the Lyapunov equation Vrabie et al. (2009). In this case, the optimal parameter of the value function $\omega^* = \mathrm{vec}(P^*)$ can be determined by solving the Ricatti equation Vrabie et al. (2009). Consider the third-order system Jiang & Jiang (2017):

$$A = \begin{bmatrix} 0 & 1 & 0 \\ 0 & 0 & 1 \\ -0.1 & -0.5 & -0.7 \end{bmatrix}, \ B = \begin{bmatrix} 0 \\ 0 \\ 1 \end{bmatrix}, \ Q = I_{3\times 3}, \ R = 1.$$

The solution of the Ricatti equation is $P^* = \begin{bmatrix} 2.36 & 2.24 & 0.90 \\ 2.24 & 4.24 & 1.89 \\ 0.90 & 1.89 & 1.60 \end{bmatrix}$. The initial policy of the PI is chosen as an admissible policy $u = -K_0 x$ with $K_0 = [0, 0, 0]$. We use the trapezoidal rule and the BQ with Matérn kernel (smoothness parameter $b = 4$) for evenly spaced samples with size $5 \le N \le 15$ to compute the PEV step of IntRL. The $l_2$-norm of the difference of the parameters $\|\hat{\omega}^{(\infty)} - \omega^*\|_2$ concerning the sample size can be shown in Figure 4, where $\hat{\omega}^{(\infty)} := \lim_{i \to \infty} \hat{\omega}^{(i)}$ is the learned parameter after sufficient iterations and $\omega^*$ is the optimal parameter.

---

[1]The code is available at https://github.com/anonymity678/Computation-Impacts-Control.git.

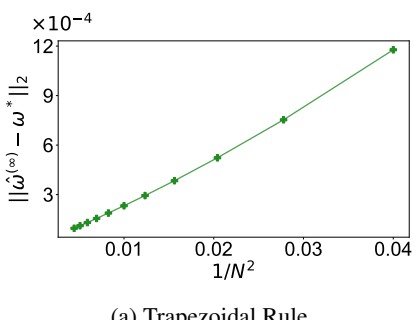

(a) Trapezoidal Rule

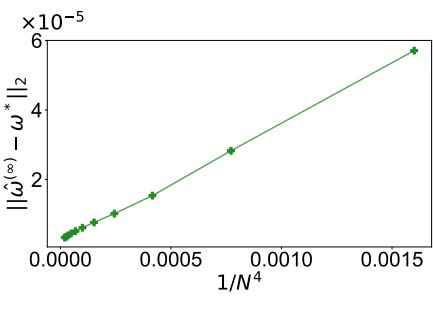

(b) BQ with Matérn Kernel

Figure 4: Simulations for Example 1. The convergence rates of the learned parameters $\hat{\omega}^{(\infty)}$ solved by the trapezoidal rule and BQ with Matérn Kernel ($b = 4$) are shown to be $O(N^{-2})$ and $O(N^{-4})$.

We demonstrate that the convergence rate of the parameter $\hat{\omega}^{(\infty)}$ aligns with the derived upper bound for the value function as outlined in Corollary 1. This observation is sufficient to validate Corollary 1 since the convergence rate of $\hat{\omega}^{(\infty)}$ inherently serves as an upper bound for the convergence rate of the value function $\hat{V}^{(\infty)}$. Specifically, we have:

$$|V^{(\infty)}(x) - V^*(x)| = \|(\hat{\omega}^{(\infty)} - \omega^*)\phi(x)\|_2 \leq \|\hat{\omega}^{(\infty)} - \omega^*\|_2 \|\phi(x)\|_2, \quad \forall x \in \Omega.$$

**Example 2 (Nonlinear System):** Then we consider a canonical nonlinear system in the original paper of IntRL (Example 1 in Vrabie & Lewis (2009)):

$$\dot{x}_1 = -x_1 + x_2, \quad \dot{x}_2 = -0.5(x_1 + x_2) + 0.5x_2 \sin^2(x_1) + \sin(x_1)u,$$

with the utility function defined as $l(x, u) = x_1^2 + x_2^2 + u^2$. The optimal value function for this system is $V^*(x) = 0.5x_1^2 + x_2^2$ and the optimal controller is $u^*(x) = -\sin(x_1)x_2$. Due to the simple formulation of the value function, we set the basis function to be $\phi(x) := x \otimes x$ as in Vrabie & Lewis (2009). The initial policy is chosen as $u^{(0)} = -1.5x_1 \sin(x_1)(x_1 + x_2)$ and we use the trapezoidal rule and the BQ with the Matérn kernel (smoothness parameter $b = 4$) for evenly spaced samples with size $5 \leq N \leq 15$ to compute the PEV step. The convergence rate of the learned parameter for Example 2 can be shown in Figure 5 and more simulation results regarding the convergence rate of the controller and the average accumulated cost can be found in Appendix J.

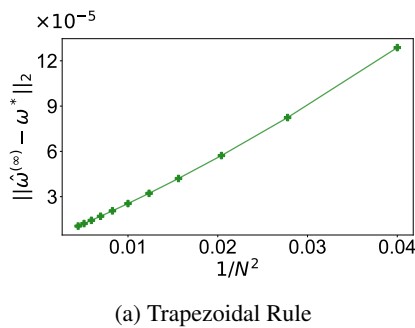

(a) Trapezoidal Rule

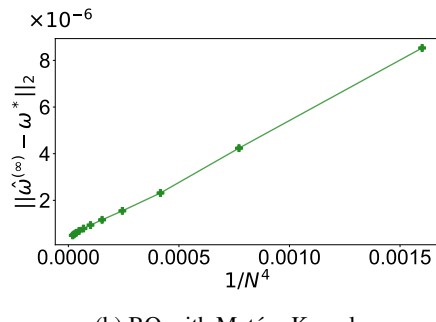

(b) BQ with Matérn Kernel

Figure 5: Simulations for Example 2. The convergence rates of the learned parameters $\hat{\omega}^{(\infty)}$ solved by the trapezoidal rule and BQ with Matérn Kernel ($b = 4$) are shown to be $O(N^{-2})$ and $O(N^{-4})$.

## 5 CONCLUSION AND DISCUSSION

This paper primarily identifies the phenomenon termed "computation impact control". Yet, it is possible to harness computational uncertainty, specifically the posterior variance in the BQ, to devise a "computation-aware" control policy. One approach is to view computational uncertainty with pessimism. In situations where computational uncertainty poses significant challenges, robust control methodologies can be employed to derive a conservative policy. Alternatively, a more optimistic perspective would embrace computational uncertainty as an opportunity for exploration.

ETHICS STATEMENT

Our research strictly adheres to ethical guidelines and does not present any of the issues related to human subjects, data set releases, harmful insights, conflicts of interest, discrimination, privacy, legal compliance, or research integrity. All experiments and methodologies were conducted in a simulation environment, ensuring no ethical or fairness concerns arise.

REPRODUCIBILITY STATEMENT

The source code associated with this paper is available for download through an anonymous GitHub repository at https://github.com/anonymity678/Computation-Impacts-Control.git. All simulation results have been uploaded to the repository, ensuring that readers and researchers can reproduce the findings presented in this paper with ease. Besides, all the theoretical proofs for the theorems and lemmas are provided in the appendix.

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

# APPENDIX

## A    MOTIVATING EXAMPLE FOR KNOWN INTERNAL DYNAMICS

The phenomenon "computation impacts control performance" also exists for the CTRL algorithm with known internal dynamics Yildiz et al. (2021). In Figure 6, we evaluate the impact of different numerical ordinary differential equation (ODE) solvers on the accumulated costs when applying the CTRL algorithm with known internal dynamics. Specifically, we consider the fixed-step Euler method with varying step sizes and the adaptive Runge-Kutta (4,5) method. Our results from the Cartpole task show that the adaptive ODE solver, in this case the Runge-Kutta (4,5) method, consistently outperforms the Euler method, yielding lower accumulated costs. Additionally, a smaller step size is associated with improved performance within the Euler method.

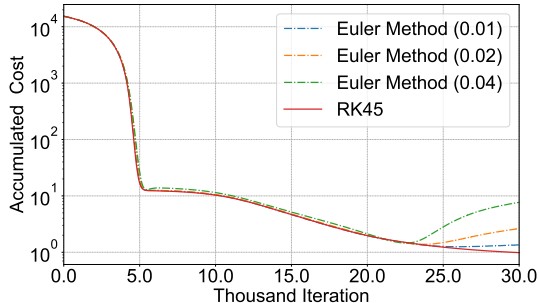

Figure 6: The illustration of the phenomenon "computation impacts control performance" when internal dynamics is known. This figure shows the accumulated costs for the Euler method with different step sizes (indicated in parentheses) and the Runge-Kutta (4,5) method in the Cartpole task with known internal dynamics.

## B    DEFINITION OF ADMISSIBLE POLICIES

**Definition 1** (Admissible Policies Vrabie & Lewis (2009)). *A control policy $u$ is admissible with respect to the cost, denoted as $u \in A(\Omega)$, if $u$ is continuous on $\Omega$, $u(0) = 0$, $u$ stabilizes the system (4) on $\Omega$, and $J(x_0, u)$ in (5) is finite for all $x_0 \in \Omega$.*

## C    CT BELLMAN EQUATION FOR KNOWN INTERNAL DYNAMICS

When the internal dynamics is considered to be *known*, which can be derived from fundamental principles such as Lagrangian or Hamiltonian mechanics, or inferred using deep learning models Chen et al. (2018), Lutter et al. (2018), and Zhong et al. (2019). In this case, the CT Bellman equation can be addressed by formulating an augmented ODE:

$$\begin{bmatrix} \dot{x}(t) \\ \dot{V}(x(t)) \end{bmatrix} = \begin{bmatrix} f(x(t)) + g(x)u(x(t)) \\ l(x(t), u(x(t))) \end{bmatrix}.$$

Here, both the state $x$ and value function $V$ act as independent variables of the augmented ODE. This can be solved by numerical ODE solvers ranging from fixed-step solvers like the Euler method to adaptive solvers such as the Runge-Kutta series Yildiz et al. (2021). The computational error for solving the CT bellman equation can be sufficiently small by choosing the adaptive ODE solvers.

## D  PROOF OF LEMMA 1

*Proof.* For $\forall V \in \mathbb{V}$ and $W \in \mathbb{V}_n \subset \mathbb{V}$, where $\mathbb{V}_n$ denotes the neighborhood of $V$. By the definition of $G(V)$ in (7), we have

$$
\begin{aligned}
&G(V + sW) - G(V) \\
&= \frac{\partial(V + sW)}{\partial x^\top} f - \frac{1}{4} \frac{\partial(V + sW)}{\partial x^\top} g R^{-1} g^\top \frac{\partial(V + sW)}{\partial x} - \frac{\partial V}{\partial x^\top} f + \frac{1}{4} \frac{\partial V}{\partial x^\top} g R^{-1} g^\top \frac{\partial V}{\partial x} \\
&= s \frac{\partial W}{\partial x^\top} f - \frac{1}{4} s^2 \frac{\partial W}{\partial x^\top} g R^{-1} g^\top \frac{\partial W}{\partial x} - \frac{1}{2} s \frac{\partial W}{\partial x^\top} g R^{-1} g^\top \frac{\partial V}{\partial x}.
\end{aligned}
$$

Thus, the Gâteaux differential Kantorovich & Akilov (2016) at $V$ is

$$
\begin{aligned}
L(W) &= G'(V)W \\
&= \lim_{s \to 0} \frac{G(V + sW) - G(V)}{s} \\
&= \frac{\partial W}{\partial x^\top} f - \frac{1}{2} \frac{\partial W}{\partial x^\top} g R^{-1} g^\top \frac{\partial V}{\partial x}.
\end{aligned}
$$

Then we will prove that $L(W)$ is continuous on $\mathbb{V}_n$. For $\forall W_0 \in \mathbb{V}_n$, we have

$$
\begin{aligned}
L(W) - L(W_0) &= \| G'(V)W - G'(V)W_0 \|_\Omega \\
&= \left\| \frac{\partial(W - W_0)}{\partial x^\top} f - \frac{1}{2} \frac{\partial(W - W_0)}{\partial x^\top} g R^{-1} g^\top \frac{\partial V}{\partial x} \right\|_\Omega \\
&\leq \left( \|f\|_\Omega + \left\| \frac{1}{2} g R^{-1} g^\top \frac{\partial V}{\partial x} \right\|_\Omega \right) \cdot \left\| \frac{\partial(W - W_0)}{\partial x} \right\|_\Omega \\
&\leq \left( \|f\|_\Omega + \left\| \frac{1}{2} g R^{-1} g^\top \frac{\partial V}{\partial x} \right\|_\Omega \right) \cdot m_1 \cdot \|W - W_0\|_\Omega,
\end{aligned}
$$

where $m_1 > 0$. Then, for $\forall \epsilon > 0$, there exists $\delta = \frac{\epsilon}{m_1 \left( \|f\|_\Omega + \| \frac{1}{2} g R^{-1} g^\top \frac{\partial V}{\partial x} \|_\Omega \right)}$ satisfying $\|L(W) - L(W_0)\|_\Omega < \epsilon$ when $\|W - W_0\|_\Omega < \delta$. Due to the continuity, $L$ is also the Fréchet derivative at $V$ Kantorovich & Akilov (2016).

Then, we prove PI defined in (8a)(8b) equals (10). From (10), we obtain

$$
G'(V^{(i)}) V^{(i+1)} = G'(V^{(i)}) V^{(i)} - G(V^{(i)}). \tag{18}
$$

According to (9) and (8b), we have

$$
\begin{aligned}
G'(V^{(i)}) V^{(i+1)} &= \frac{\partial V^{(i+1)}}{\partial x^\top} f - \frac{1}{2} \frac{\partial V^{(i+1)}}{\partial x^\top} g R^{-1} g^\top \frac{\partial V^{(i)}}{\partial x} \\
&= \frac{\partial V^{(i+1)}}{\partial x^\top} \left( f + g u^{(i+1)} \right),
\end{aligned} \tag{19a}
$$

$$
\begin{aligned}
G'(V^{(i)}) V^{(i)} &= \frac{\partial V^{(i)}}{\partial x^\top} f - \frac{1}{2} \frac{\partial V^{(i)}}{\partial x^\top} g R^{-1} g^\top \frac{\partial V^{(i)}}{\partial x} \\
&= \frac{\partial V^{(i)}}{\partial x^\top} f - 2 u^{(i+1)^\top} R u^{(i+1)},
\end{aligned} \tag{19b}
$$

$$
\begin{aligned}
G(V^{(i)}) &= \frac{\partial V^{(i)}}{\partial x^\top} f + q - \frac{1}{4} \frac{\partial V^{(i)}}{\partial x^\top} g R^{-1} g^\top \frac{\partial V^{(i)}}{\partial x} \\
&= \frac{\partial V^{(i)}}{\partial x^\top} f + q - u^{(i+1)^\top} R u^{(i+1)}.
\end{aligned} \tag{19c}
$$

Pluging (19) into (18), we obtain

$$
\frac{\partial V^{(i+1)}}{\partial x^\top} \left( f + g u^{(i+1)} \right) = -q - u^{(i+1)^\top} R u^{(i+1)}.
$$

Which means

$$
\dot{V}^{(i+1)} = -q - u^{(i+1)^\top} R u^{(i+1)}. \tag{20}
$$

Finally, (8a) can be obtained by integrating (20). $\qquad\square$

## E    PROOF OF THEOREM 2

First, we introduce a useful lemma for the convergence of the standard Newton's method (without computational error).

**Lemma 2** (Convergence of the Standard Newton's Method Ostrowski (2016); Lancaster (1966)). *Suppose these exists an $V^{(0)} \in \mathbb{V}$ for which $\left[G'(V^{(0)})\right]^{-1}$ exists and $G$ is twice Fréchet differentiable on $B_0$, where $B_0 := \{V \in \mathbb{V} : \|V - V^{(0)}\|_{\Omega} \leq r_0\}$. Define $r_0 := \|V^{(1)} - V^{(0)}\|_{\Omega} = \|\left[G'(V^{(0)})\right]^{-1} G(V^{(0)})\|_{\Omega}$ and $L_0 := \sup_{V \in B_0} \|\left[G'(V^{(0)})\right]^{-1} G''(V)\|_{\Omega}$, if we have $r_0 L_0 \leq \frac{1}{2}$, then the sequence $V^{(i)} \in B_0$ generated by Newton's method converges to the unique solution of $G(V) = 0$, i.e., $V^*$ in $B_0$. Moreover, we have $\|V^{(i)} - V^*\|_{\Omega} \leq \frac{2^{-i}(2r_0 L_0)^{2^i}}{L_0}$.*

Then we will analyze the convergence property of Newton's method for solving $G(V) = 0$ with an extra error term $E^{(i)} \leq \bar{\epsilon}$ in (11). We first study the iteration error between the value function for standard Newton's method $V^{(i)}$ and the value function for Newton's method incorporated with an extra error term $\hat{V}^{(i)}$. Specifically, we have the subsequent proposition:

**Proposition 1** (Iteration Error for Newton's Method). *If $\hat{V}^{(i)} \in B$, $G''$ exists and continuous in $B$, where $B := \{V \in \mathbb{V} : \|V - V^{(0)}\|_{\Omega} \leq r_0 + d\}$ with $d \geq 0$. If $G'(\hat{V}^{(i)})$ has an inverse then*

$$V^{(i+1)} - \hat{V}^{(i+1)} = \left[G'(\hat{V}^{(i)})\right]^{-1} \left\{ \int_{V^{(i)}}^{\hat{V}^{(i)}} G''(V^{(i)} - V, \cdot) \, \mathrm{d}V + \int_{V^{(i)}}^{\hat{V}^{(i)}} G''(h_i, \cdot) \, \mathrm{d}V \right\} - E^{(i)},$$

(21)

*where $h_i = -\left[G'(V^{(i)})\right]^{-1} G(V^{(i)})$.*

*Proof.* Subtract (10) with (11), we have

$$V^{(i+1)} - \hat{V}^{(i+1)} = V^{(i)} - \hat{V}^{(i)} - \left[G'(V^{(i)})\right]^{-1} G(V^{(i)}) + \left[G'(\hat{V}^{(i)})\right]^{-1} G(\hat{V}^{(i)}) - E^{(i)}$$

$$= \left[G'(\hat{V}^{(i)})\right]^{-1} \left[G(\hat{V}^{(i)}) + G'(\hat{V}^{(i)})(V^{(i)} - \hat{V}^{(i)}) + G'(\hat{V}^{(i)})h_i\right] - E^{(i)}.$$

(22)

Besides, we notice that

$$G(\hat{V}^{(i)}) = G(V^{(i)}) + G'(V^{(i)})(\hat{V}^{(i)} - V^{(i)}) + \int_{V^{(i)}}^{\hat{V}^{(i)}} G''(\hat{V}^{(i)} - V, \cdot) \, \mathrm{d}V,$$

$$G'(\hat{V}^{(i)}) = G'(V^{(i)}) + \int_{V^{(i)}}^{\hat{V}^{(i)}} G''(V) \, \mathrm{d}V,$$

(23)

$$G'(\hat{V}^{(i)})h_i = -G(V^{(i)}) + \int_{V^{(i)}}^{\hat{V}^{(i)}} G''(h_i, \cdot) \, \mathrm{d}V.$$

By substituting (23) into (22), we can obtain (21). $\square$

Then, we will use Proposition 1 to achieve the bound of the iteration error $\|V^{(i)} - \hat{V}^{(i)}\|_{\Omega}$, as summarized in the subsequent lemma.

**Lemma 3** (Iteration Error Bound for Newton's Method). *Under the Assumption in Lemma 2 and additionally assume that $G''$ exists and is continuous in $B$ and $G'(V)$ is nonsingular for $\forall V \in B$. If there exists a number $\Phi, M > 0$, such that $\frac{1}{\Phi} \geq \sup_{V \in B} \|[G'(V)]^{-1}\|_{\Omega}$, $M \geq \sup_{V \in B} \|G''(V)\|_{\Omega}$ and $\Phi - Mr_0 > \sqrt{2M\bar{\epsilon}\Phi}$ and $d > \frac{2\bar{\epsilon}\Phi}{\Phi - Mr_0}$. We have $\hat{V}^{(i)} \in B$ and*

$$\|V^{(i)} - \hat{V}^{(i)}\|_{\Omega} \leq \frac{2\bar{\epsilon}\Phi}{\Phi - Mr_0}.$$

(24)

*Proof.* The Lemma is proved by induction. Recall that $V^{(0)} = \hat{V}^{(0)}$, we obtain

$$
\begin{aligned}
\|V^{(1)} - \hat{V}^{(1)}\|_\Omega &= \|E^{(0)}\|_\Omega \\
&\leq \bar{\epsilon} \\
&\leq \frac{\bar{\epsilon}\Phi}{\Phi - Mr_0} \\
&\leq \frac{2\bar{\epsilon}\Phi}{\Phi - Mr_0}.
\end{aligned}
$$

Thus (24) holds for $i = 1$. Now suppose (24) is proved for $i$. Because $d > \frac{2\bar{\epsilon}\Phi}{\Phi - Mr_0}$ and $V^{(i)} \in B_0$ according to Lemma 2, we have $\hat{V}^{(i)} \in B$. Then by setting $V = V^{(i)} + \tau(\hat{V}^{(i)} - V^{(i)}) \in B$, we get

$$
\left\| \int_{V^{(i)}}^{\hat{V}^{(i)}} G''(V^{(i)} - V, \cdot)\, dV \right\|_\Omega \leq \|V^{(i)} - \hat{V}^{(i)}\|_\Omega^2 \left\| \int_0^1 \tau G''(V^{(i)} + \tau(\hat{V}^{(i)} - V^{(i)}))\, d\tau \right\|_\Omega
$$

$$
\leq \frac{1}{2} M \|V^{(i)} - \hat{V}^{(i)}\|_\Omega^2.
$$

Denoting $\delta = \frac{2\bar{\epsilon}\Phi}{\Phi - Mr_0}$, by (21), we have

$$
\begin{aligned}
&\|V^{(i+1)} - \hat{V}^{(i+1)}\|_\Omega \\
&= \left\| \left[ G'(\hat{V}^{(i)}) \right]^{-1} \left\{ \int_{V^{(i)}}^{\hat{V}^{(i)}} G''(V^{(i)} - V, \cdot)\, dV + \int_{V^{(i)}}^{\hat{V}^{(i)}} G''(h_i, \cdot)\, dV \right\} - E^{(i)} \right\|_\Omega \\
&\leq \left\| \left[ G'(\hat{V}^{(i)}) \right]^{-1} \right\|_\Omega \left\{ \left\| \int_{V^{(i)}}^{\hat{V}^{(i)}} G''(V^{(i)} - V, \cdot)\, dV \right\|_\Omega + \left\| \int_{V^{(i)}}^{\hat{V}^{(i)}} G''(h_i, \cdot)\, dV \right\|_\Omega + \bar{\epsilon} \right\} \\
&\leq \frac{1}{\Phi} \left\{ \frac{1}{2} M\delta^2 + M\|h_i\|_\Omega \delta + \bar{\epsilon} \right\}.
\end{aligned}
$$

According to Lemma 2, $\|h_i\|_\Omega < r_0$, $i = 1, 2, ...$, so we have

$$
\begin{aligned}
\left\| V^{(i+1)} - \hat{V}^{(i+1)} \right\|_\Omega &\leq \frac{M\delta^2 + 2M\|h_i\|_\Omega \delta + 2\Phi\bar{\epsilon}}{2\Phi} \\
&\leq \frac{\delta}{2} \frac{M\delta + 2Mr_0 + (\Phi - Mr_0)}{\Phi} \\
&= \frac{\delta}{2} \frac{M\delta + Mr_0 + \Phi}{\Phi}.
\end{aligned}
$$

The condition $\Phi - Mr_0 > \sqrt{2M\bar{\epsilon}\Phi}$ implies that

$$
M\delta + Mr_0 < \Phi.
$$

Hence

$$
\|V^{(i+1)} - \hat{V}^{(i+1)}\|_\Omega \leq \delta = \frac{2\bar{\epsilon}\Phi}{\Phi - Mr_0}.
$$

This completes the proof of Lemma 3. $\qquad\square$

Using a direct combination of Lemma 2 and Lemma 3, and perform triangle inequality in Banach space, we can obtain Theorem 2.

## F    DETAILS FOR MATÉRN KERNEL

We focus on scale-dependent Matérn kernels, which are radial basis kernels with Fourier decay characterized by a smoothness parameter $b > 1/2$ for samples belonging to $\mathbb{R}$ Wendland (2004); Pagliana et al. (2020):

$$
K_{\text{Matérn}}(x, x') = \left( \frac{\|x - x'\|}{\rho} \right)^{b - \frac{1}{2}} \beta_{b - \frac{1}{2}} \left( \frac{\|x - x'\|}{\rho} \right),
$$

where $\|x - x'\|$ is the Euclidean distance between $x$ and $x'$, $b$ is a parameter that controls the smoothness of the function, $\rho$ is the length scale parameter and $\beta_\alpha$ is the modified Bessel function of the second kind with parameter $\alpha$ Seeger (2004). A specific and important property of the Matérn kernel is that for $b > \frac{1}{2}$, the RKHS corresponding to different scales are all equivalent to the Sobolev space $W_2^b$ Seeger (2004); Matérn (2013); Kanagawa et al. (2020).

## G    ILLUSTRATION OF BQ

Here, we will take the integral

$$\int_2^{10} \left[ \frac{t}{10} \sin\left(\frac{3\pi}{5}t\right) + 2 \right] \, \mathrm{d}t \tag{25}$$

as an example to illustrate BQ using Wiener kernel and Matérn kernel, see Figure 7 and 8 respectively. It can be seen from the figures that BQ can be understood as first fitting the integrand using Gaussian process regression, and then performing integration. The accuracy of BQ is related to the choice of the Gaussian process kernel and the sample size. For evenly spaced samples, as the sample size increases, the accuracy of BQ improves. Besides, we also plot the integral values and their computational errors for further illustration; see figure 9.

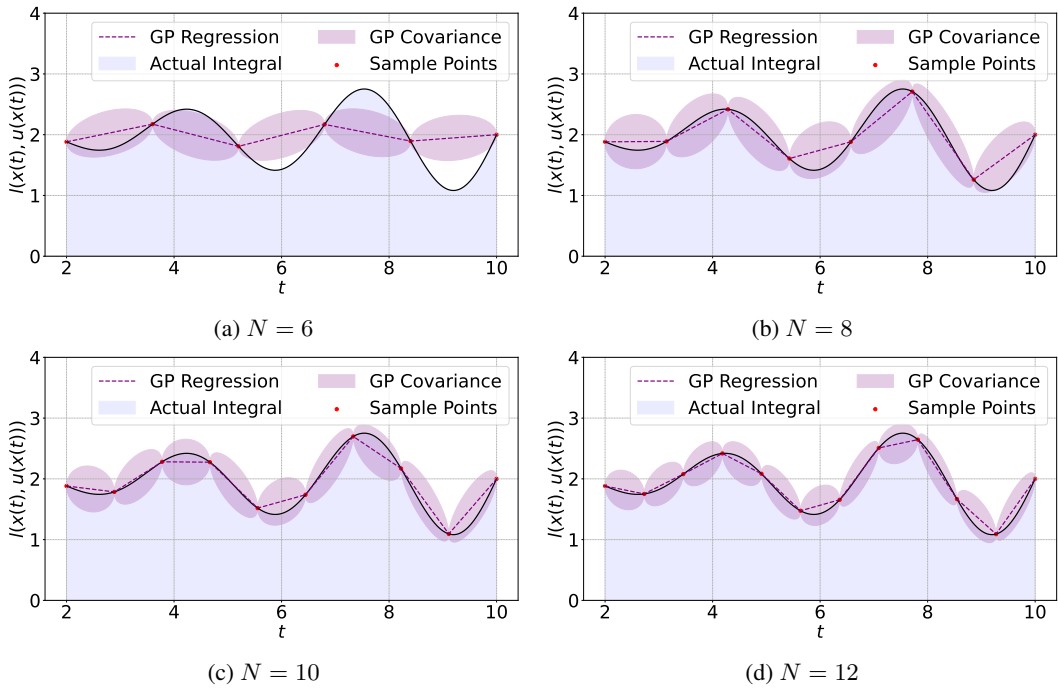

(a) $N = 6$
(b) $N = 8$
(c) $N = 10$
(d) $N = 12$

Figure 7: Illustration of BQ with Wiener kernel (equal to the trapezoidal rule) for the integral in (25).

## H    DISCUSSION ABOUT FULL COLUMN RANK CONDITION

**Lemma 4** (Full Column Rank of $\Theta^{(i)}$ Vrabie & Lewis (2009))**.** *Suppose $u^{(i)}(x) \in A(\Omega)$, and the set $\{\phi_k\}_1^{n_\phi}$ is linearly independent. Then, $\exists T > 0$ such that for $\forall x(t)$ generated by $u^{(i)}(x)$, $\{\phi_k(x(t+T)) - \phi_k(x(t))\}_1^{n_\phi}$ is also linear independent.*

Based on this lemma, there exist values of $T_1, T_2, ..., T_{m+1}$ such that $\Theta^{(i)}$ is invertible and thus (15) has unique solution.

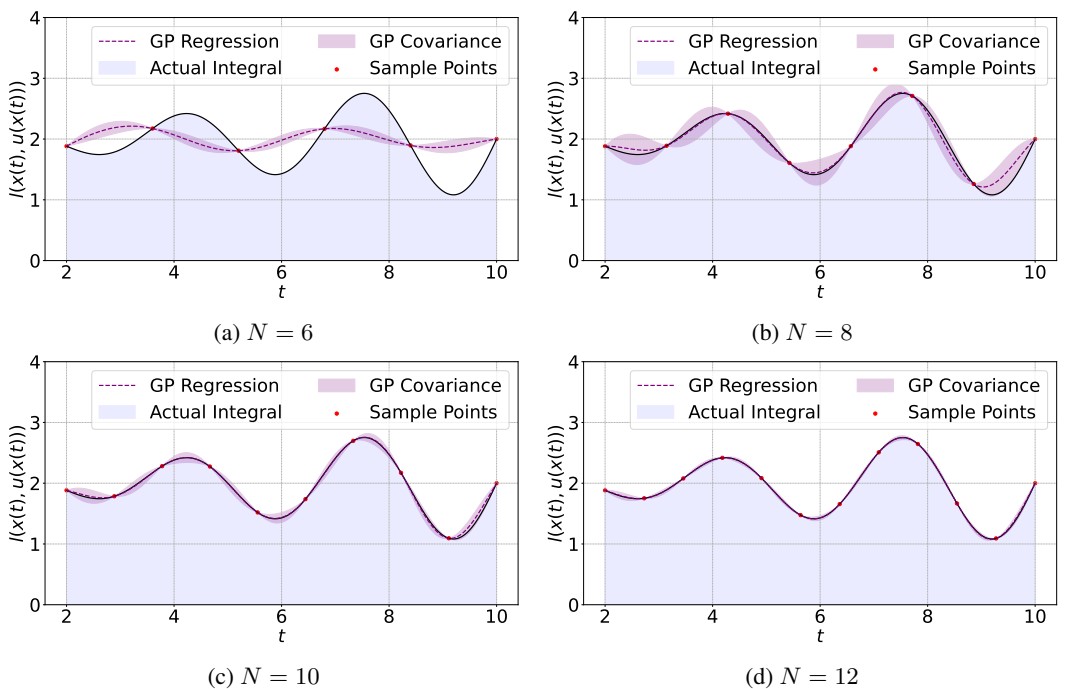

Figure 8: Illustration of BQ with Matérn kernel for the integral in (25).

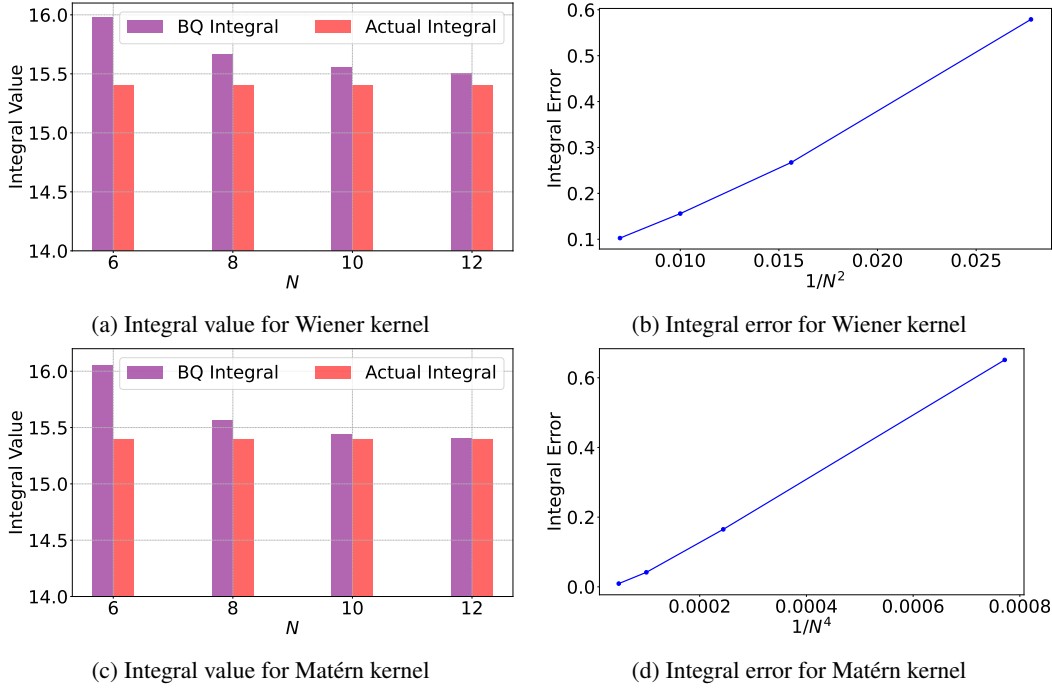

Figure 9: Integral values and errors of BQ for the integral in (25).

# I  PROOF OF THEOREM 3

First, we will give a proposition which presents the upper bound of the error between $\omega^{(i)}$ and $\hat{\omega}^{(i)}$ if the same control policy $\hat{u}^{(i)}$ is utilized.

**Proposition 2** (Solution Error of PEV). *For the same control policy $\hat{u}^{(i)}$, the difference of the solution of the (15) and (16) can be bounded by:*

$$\|\omega^{(i)} - \hat{\omega}^{(i)}\|_2 \leq \|(\hat{\Theta}^{(i)\top}\hat{\Theta}^{(i)})^{-1}\hat{\Theta}^{(i)\top}\|_2 \|\delta\Xi^{(i)}\|_2.$$

*Proof.* For the same control policy $u^{(i)}$, we have $\hat{\Theta}^{(i)} = \Theta^{(i)}$. Consider the linear matrix equation (15) and (16), we have

$$\begin{aligned}
\|\omega^{(i)} - \hat{\omega}^{(i)}\|_2 &= \|(\hat{\Theta}^{(i)\top}\hat{\Theta}^{(i)})^{-1}\hat{\Theta}^{(i)\top}(\Xi^{(i)} - \hat{\Xi}^{(i)})\|_2 \\
&\leq \|(\hat{\Theta}^{(i)\top}\hat{\Theta}^{(i)})^{-1}\hat{\Theta}^{(i)\top}\|_2 \|\Xi^{(i)} - \hat{\Xi}^{(i)}\|_2 \\
&\leq \|(\hat{\Theta}^{(i)\top}\hat{\Theta}^{(i)})^{-1}\hat{\Theta}^{(i)\top}\|_2 \|\delta\Xi^{(i)}\|_2.
\end{aligned}$$

In this equation, $\delta\Xi^{(i)}$ is a vector with its $k^{\text{th}}$ row defined as $\delta\xi(T_k, T_{k+1}, \hat{u}^{(i)})$. Note that $\delta\xi(T_k, T_{k+1}, \hat{u}^{(i)})$ serves as an upper bound for the computational error of the integral $\xi(T_k, T_{k+1}, \hat{u}^{(i)}) = \int_{T_k}^{T_{k+1}} l(x(s), \hat{u}^{(i)}(x(s)))\, ds$. $\qquad \square$

Then we present the proof of Theorem 3:

*Proof.* We choose the norm in the Banach space $\mathbb{V}$ as the infinity norm, i.e., $\|V\|_\Omega = \|V\|_\infty = \sup_{x \in \Omega} |V(x)|$. Because $\Omega$ is bounded and $\phi_k \in \mathbb{C}^1, \forall k = 1, 2, ..., n_\phi$, thus $\|\phi_k\|_\infty = \sup_{x \in \Omega} |\phi_k(x)|$ and $\|\phi\|_\infty := \max_{1 \leq k \leq n_\phi}\{\|\phi_k\|_\infty\}$ exists. From Proposition 2 and Cauchy-Schwarz inequality, we have

$$\begin{aligned}
\left|E^{(i)}(x)\right| &= \|E^{(i)}(x)\|_2 \\
&\leq \|\hat{\omega}^{(i)} - \omega^{(i)}\|_2 \|\phi(x)\|_2 \\
&\leq \|(\hat{\Theta}^{(i)\top}\hat{\Theta}^{(i)})^{-1}\hat{\Theta}^{(i)\top}\|_2 \|\delta\Xi^{(i)}\|_2 \|\phi(x)\|_2 \\
&\leq \sup_i \left\{\|(\hat{\Theta}^{(i)\top}\hat{\Theta}^{(i)})^{-1}\hat{\Theta}^{(i)\top}\|_2 \|\delta\Xi^{(i)}\|_2\right\} \|\phi(x)\|_2, \quad \forall x \in \Omega.
\end{aligned}$$

Thus, we have

$$\begin{aligned}
\|E^{(i)}\|_\infty &= \sup_x |E^{(i)}(x)| \\
&\leq \sup_i \left\{\|(\hat{\Theta}^{(i)\top}\hat{\Theta}^{(i)})^{-1}\hat{\Theta}^{(i)\top}\|_2 \|\delta\Xi^{(i)}\|_2\right\} \|\phi\|_\infty \qquad (26) \\
&= \bar{\epsilon}.
\end{aligned}$$

Combining (26) with (12) in Theorem 2, we can obtain

$$\|\hat{V}^{(i)} - V^*\|_\infty \leq \frac{2\Phi\bar{\epsilon}}{\Phi - Mr_0} + \frac{2^{-i}(2r_0 L_0)^{2^i}}{L_0},$$

which directly leads to

$$|\hat{V}^{(i)}(x) - V^*(x)| \leq \frac{2\Phi\bar{\epsilon}}{\Phi - Mr_0} + \frac{2^{-i}(2r_0 L_0)^{2^i}}{L_0}, \quad \forall x \in \Omega.$$

$$\square$$

## J  ADDITIONAL SIMULATION RESULTS

In this subsection, we show additional simulations results for the convergence rate of the controller and the accumulated costs for Example 1 and Example 2 in Section 4.

**Example 1:**

**Control Gain Matrix:** For linear systems, the learned control policy adheres to $u^{(\infty)}(x) = -\hat{K}^{(\infty)}x$ where $\hat{K}^{(\infty)}$ is the learned control gain matrix which is computed as Jiang & Jiang (2017):

$$\hat{K}^{(\infty)} = R^{-1}B^\top \hat{P}^{(\infty)}.$$

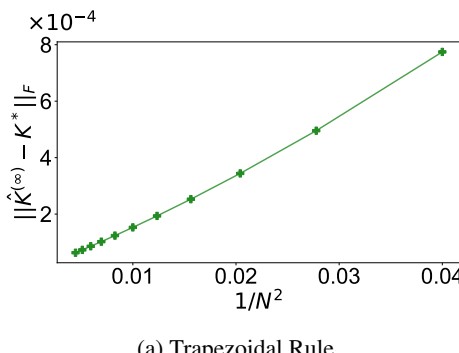 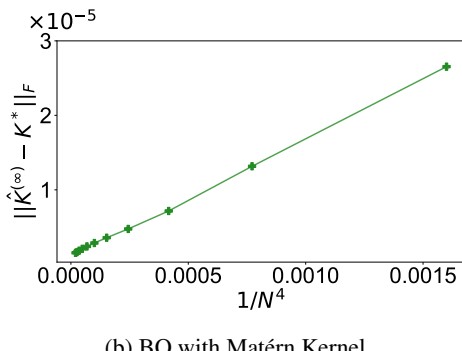

(a) Trapezoidal Rule                    (b) BQ with Matérn Kernel

Figure 10: Simulations for Example 1 showing convergence rates of $\hat{K}^{(\infty)}$ computed via the trapezoidal rule and BQ with Matérn Kernel ($b = 4$) as $O(N^{-2})$ and $O(N^{-4})$ respectively.

Here, $\hat{P}^{(\infty)}$ ensures $\hat{\omega}^{(\infty)} = \text{vec}(\hat{P}^{(\infty)})$. The Frobenius norm difference between the learned and optimal control gain matrix $K^* = \begin{bmatrix} 0.90 & 1.89 & 1.60 \end{bmatrix}$, denoted as $\|\hat{K}^{(\infty)} - K^*\|_F$, is depicted in Figure 10.

**Average Accumulated Cost:** The average accumulated costs of the learned and optimal policies, represented as $J$ and $J^*$ respectively, with initial state $x_0 \sim \mathcal{N}(0, 100^2 \cdot I_{3\times3})$, are defined as:

$$J = \mathbb{E}_{x_0 \sim \mathcal{N}(0,100^2 \cdot I_{3\times3})} \left\{ \hat{V}^{(\infty)}(x_0) \right\}, \quad J^* = \mathbb{E}_{x_0 \sim \mathcal{N}(0,100^2 \cdot I_{3\times3})} \left\{ V^*(x_0) \right\}.$$

Their difference concerning the sample size $N$ is shown in Figure 11.

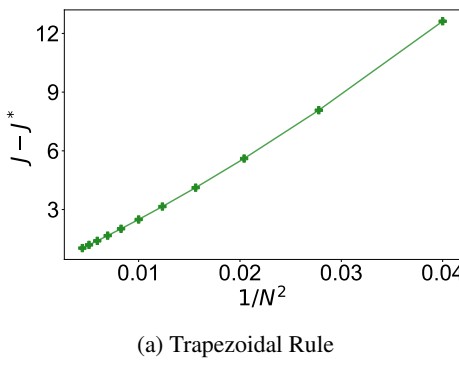 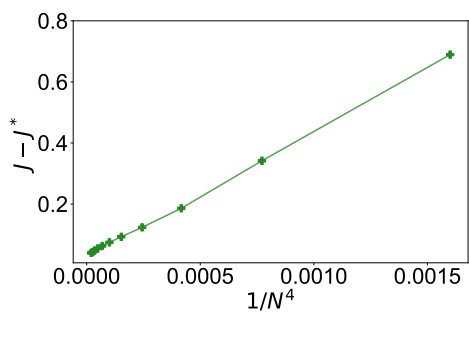

(a) Trapezoidal Rule                    (b) BQ with Matérn Kernel

Figure 11: Simulations for Example 1 illustrating convergence rates of the average accumulated cost $J$ computed via the trapezoidal rule and BQ with Matérn Kernel ($b = 4$) as $O(N^{-2})$ and $O(N^{-4})$ respectively.

**Example 2:**

**Control Policy:** For nonlinear systems, the control gain matrix does not exist. Instead, the learned control policy is given by

$$\hat{u}^{(\infty)}(x) = -\frac{1}{2} R^{-1} g(x)^\top \nabla_x \hat{V}^{(\infty)}.$$

Therefore, we present the average control difference across state $x \sim \mathcal{N}(0, 100^2 \cdot I_{2\times2})$:

$$\mathbb{E}_{x \sim \mathcal{N}(0,100^2 \cdot I_{2\times2})} \left\{ |\hat{u}^{(\infty)}(x) - u^*(x)| \right\},$$

where $u^*(x)$ represents the optimal controller. The average difference between the learned and optimal control policies is presented in Figure 12.

**Average Accumulated Cost:** The average accumulated costs of the learned and optimal policies, represented as $J$ and $J^*$ respectively, with initial state $x_0 \sim \mathcal{N}(0, 100^2 \cdot I_{2\times2})$, are defined as:

$$J = \mathbb{E}_{x_0 \sim \mathcal{N}(0,100^2 \cdot I_{2\times2})} \left\{ \hat{V}^{(\infty)}(x_0) \right\}, \quad J^* = \mathbb{E}_{x_0 \sim \mathcal{N}(0,100^2 \cdot I_{2\times2})} \left\{ V^*(x_0) \right\}.$$

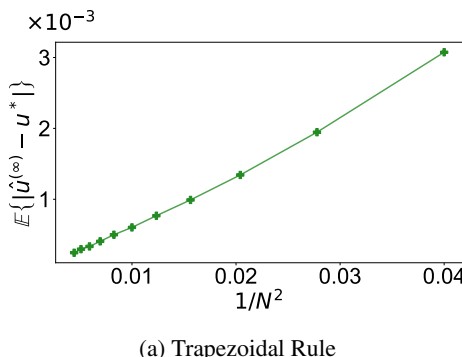 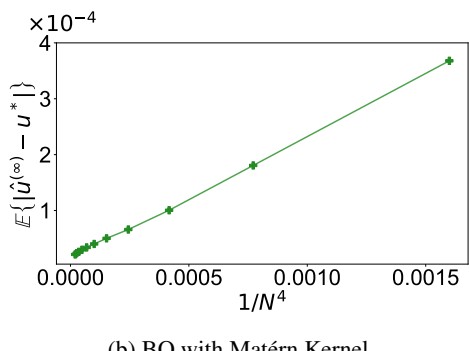

(a) Trapezoidal Rule        (b) BQ with Matérn Kernel

Figure 12: Simulations for Example 2 displaying convergence rates of $\hat{u}^{(\infty)}$ via the trapezoidal rule and BQ with Matérn Kernel ($b = 4$) as $O(N^{-2})$ and $O(N^{-4})$ respectively.

Their difference concerning the sample size $N$ is shown in Figure 13.

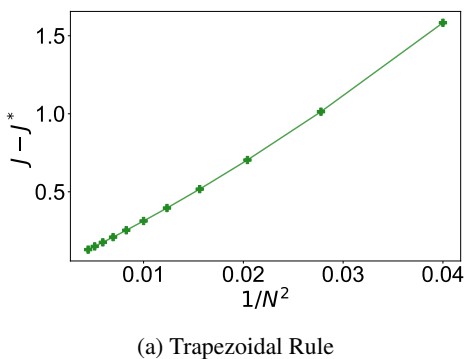 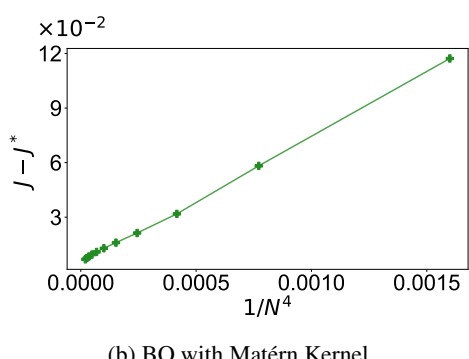

(a) Trapezoidal Rule        (b) BQ with Matérn Kernel

Figure 13: Simulations for Example 2 illustrating convergence rates of the average accumulated cost $J$ via the trapezoidal rule and BQ with Matérn Kernel ($b = 4$) as $O(N^{-2})$ and $O(N^{-4})$ respectively.

## K   DISCUSSION ABOUT THE LIMITATIONS OF INTRL

In this appendix, we want to discuss the limitations of IntRL algorithm. IntRL is a canonical control task in CTRL, just like the Q learning in DTRL. However, IntRL is still underdeveloped for high-dimensional systems compared to Q learning. As shown in Vrabie & Lewis (2009); Modares et al. (2014); Wallace & Si (2023), IntRL is typically applied to systems with no more than 4 dimensions. This is because IntRL is hard to converge in high-dimensional systems. The difficulties in achieving convergence in high-dimensional systems can be attributed to several factors:

- **Limited Approximation Capability:** IntRL uses a linear combination of basis functions to approximate the value function, which is less powerful than the neural network-based approaches prevalent in DTRL. Besides, in Wallace & Si (2023), the author observes that the condition number of the pseudo-inverted matrices for IntRL will degrade significantly due to an additional basis function. This conditioning issue limits IntRL to leverage complex basis functions to obtain larger approximation capability. It is emphasised in Wallace & Si (2023) that severe numerical breakdowns to even small increments in problem dimension.

- **Lack of Effective Exploration Mechanisms:** IntRL struggles with poor data distribution due to inadequate exploration strategies. Common DTRL methods improve exploration by adding noise during data collection, but in continuous systems, this leads to complex stochastic differential equations (SDEs) that are challenging to manage. In Wallace & Si (2023), the author emphasises that IRL's lack of exploration noise causes data quality degradation especially when the state is regulated to the origin. Besides, it is observed that a lack of exploration noise will result in the phenomenon of "hyperparameter deadlock".

- **Absence of Advanced Training Techniques:** Techniques like replay buffers, parallel exploration, delayed policy updates, etc., which enhance sample efficiency and training in DTRL, are lacking in IntRL.
- **No Discount Factor:** The absence of a discount factor in IntRL makes it difficult to ensure that policy iteration acts as a contraction mapping, which can affect the stability of algorithm's training process.
- **Non-existence of Q function:** In CT systems, the concept of a Q function is not directly applicable. The absence of Q function impedes the direct translation of many DTRL methodologies and insights to IntRL.

Considering these challenges, CTRL algorithms should be investigated and developed for complex and high-dimensional scenarios in future research.

## L  FURTHER DISCUSSION ABOUT THE MOTIVATION OF CTRL

Many physical and biological systems are inherently continuous in time, governed by ODEs. Compared to discretising time and then applying DTRL algorithms, directly employing CTRL offers these advantages Jiang & Jiang (2017); Wallace & Si (2023):

- **Smoother Control Law**: Direct application of CTRL typically results in smoother control outputs. This contrasts with the outcomes of coarse discretization, where control is less smooth and can lead to suboptimal performance Doya (2000).
- **Time Partitioning**: When addressing CTRL directly, there's no need to predefine time partitioning. Instead, it is efficiently managed by numerical integration algorithms, which find the appropriate granularity Yildiz et al. (2021). Thus, CTRL is often a better choice when the time interval is uneven.
- **Enhanced Precision with CT Transition Models**: Employing CT transition models in modelling CT systems offers a higher degree of precision compared to DT transition models. As evidenced in Wallace & Si (2023), comparisons between CT and DT trajectories, such as in the CartPole and Acrobat tasks, reveal that CT models align more closely with true solutions, especially in scenarios involving irregularly sampled data.

While direct comparisons of performance between DTRL and CTRL in literature are rare, CTRL's prominence in the financial sector, especially in portfolio management, is notable. Many portfolio management models inherently rely on Stochastic Differential Equations (SDEs). A notable example includes the comparison in Wang & Zhou (2020) between the EMV algorithm (a CTRL-based approach) and DDPG (a DTRL approach) Lillicrap et al. (2015). This comparison showed the EMV algorithm's superior performance, highlighting CTRL's advantages in scenarios that utilize SDE-based models.

The distinct benefits of CTRL across various domains, from its enhanced precision in modeling to its adaptability in handling irregular time intervals, establish it as a vital and influential methodology. Its applicability in diverse sectors, notably in complex and dynamic fields like engineering and finance, make CTRL an increasingly relevant and powerful tool.

