# OpenReview forum: "Impact of Computation in Integral Reinforcement Learning for Continuous-Time Control"
_ICLR.cc/2024/Conference — ICLR 2024 spotlight_

### Official Review · Reviewer_EcSW · 2023-10-28

**Soundness:** 4 excellent
**Presentation:** 2 fair
**Contribution:** 3 good
**Rating:** 8
**Confidence:** 3

**Summary:**

This work investigates the effects of choice of quadrature rules for integral RL; especially when the true dynamics is unknown.
It discusses how the computational error in policy evaluation stage affects each iteration of PI, which is shown to be corresponding to the Newton’s method: theoretically, the work proves the local convergence rates for IntRL; and the findings are validated by some control tasks.
Furthermore, the work shows that the case where the utility function lives in an RKHS as corollary.

**Strengths:**

1. Conceptually this work may bring up a new research direction of studying the effects of “approximation error” for ODE or problems including integrals in ML field (e.g. for neural ODE).  This is the point I particularly find value in this work.
2. The claims are validated by simple yet informative experiments.

**Weaknesses:**

1. While CT formulation helps in some analysis or for some applications, it was a bit unclear what the motivations behind studying CTRL if the task can be done with DT formulations.
(Especially when the time interval is even; in which case DT can well manage.)
In particular, for DT and CT systems, there should be different conditions for the solutions to exist.
For contact-rich dynamics for example, this kind of analysis becomes harder for example.
(Does PIM ensures the existence of solutions throughout the whole process?)
Also, for stochastic systems, CT requires more conditions for certain analysis.
2. About approximation error of value functions; if we know the utility lives in certain RKHS, can we say anything about the value function which may validate the assumptions?  At least, there should be a trivial case for this assumption: if you know the value function exactly, that becomes a single basis function; when is it continuously differentiable?  A bit more discussions needed.
(also there is a type (additional “]”) for the interval for the integral value.)
3. In Appendix H; no approach to find a suitable T?  There may require some discussions on how rare the independence property fails for a random T.

**Questions:**

1. For Appendix G, not only for the figure for the utility itself, a plot for the integral of the utility and the worst case error would be informative too.
2. Are there any answer to the weakness points?

---

> ### Author Response · Authors · 2023-11-18
> **Response to Reviewer EcSW Part1**
>
> Thank you for acknowledging the insights in our paper and for your valuable comments.
>
> **[Q1: The motivations behind studying CTRL are unclear]**
>
> **Response:**
>
> Many physical and biological systems are inherently continuous in time, governed by ordinary differential equations (ODEs). Compared to discretising time and then applying discrete-time (DT) RL algorithms, directly employing CTRL offers these advantages [1]:
>
> 1. **Smoother Control Law**: Direct application of CTRL typically results in smoother control outputs. This contrasts with the outcomes of coarse discretization, where control is less smooth and can lead to suboptimal performance [2].
> 2. **Time Partitioning**: When addressing CTRL directly, there's no need to predefine time partitioning. Instead, it is efficiently managed by numerical integration algorithms, which find the appropriate granularity [3]. Thus, CTRL is often a better choice when the time interval is uneven.
>
>
> In light of these factors, we believe that studying CTRL is crucial.
>
> **[Q2: For DT and CT systems, there should be different conditions for the solutions to exist. Does PIM ensure the existence of solutions throughout the whole process?]**
>
> **Response:**
>
> We are not sure about what the word "solution" refers to. So, we answer this question based on two circumstances:
>
> **(1) If "solution" refers to the solution of ODE, i.e., the trajectory of the state**
>
> This is indeed an important question.
>
> First, we want to highlight that our algorithm doesn't directly solve the ODE. Instead, it approximates the value function using *data* (state samples) directly in the policy evaluation (PEV) step in each iteration.
>
> Second, for continuous time-invariant systems, the Lipschitz continuity of the closed-loop systems can guarantee the existence of the solution. For the control affine system (4) in our paper, if the functions $f$ and $g$ are Lipschitz continuous, and PIM in each iteration yields a continuous control policy $u^{(i)}$, then $f(x)+g(x)u^{(i)}$ will also be Lipschitz continuous. In this case, the existence of solution of the closed-loop system can be guaranteed.
>
> **(2) If "solution" refers to the solution of the algorithm, i.e., the optimal controller**
>
> DT and CT systems have different conditions for the optimality of solutions. DT systems focus on the Bellman optimality equation, while CT systems center on the Hamilton–Jacobi–Bellman (HJB) equation. The table below summarizes these conditions:
>
> | System |                     Optimality Condition                     |          Condition of PEV           |
> | :----: | :----------------------------------------------------------: | :---------------------------------: |
> |   DT   | Bellman optimality equation: $V^*(k)=\min_u\\left\\{l(x(k),u(k))+V^*(x(k+1))\\right\\}$ |       Bellman equation: (1a）       |
> |   CT   | HJB equation: $\min_u\\left\\{\dot{V}^*(x(t))+l(x(t), u(t))\\right\\}=0$; Equal to (7) for control affine systems | (Nonlinear) Lyapunov function: (1b) |
>
> In ideal conditions without learning errors or computational errors, iterative policy iteration in CT systems will converge to the optimal control solution, given an initial admissible policy [4]. In the presence of computational error, the convergence analysis of the policy iteration is addressed in our paper's Theorem 2.
>
> **References**
>
> [1] Yu Jiang and Zhong-Ping Jiang. Robust adaptive dynamic programming. John Wiley & Sons, 2017.
>
> [2] Kenji Doya. Reinforcement learning in continuous time and space. Neural Computation, 12(1): 219–245, 2000
>
> [3] Cagatay Yildiz, Markus Heinonen, and Harri L ̈ahdesm ̈aki. Continuous-time model-based reinforcement learning. In International Conference on Machine Learning, pp. 12009–12018. PMLR, 2021.6
>
> [4] Draguna Vrabie and Frank Lewis. Neural network approach to continuous-time direct adaptive optimal control for partially unknown nonlinear systems. Neural Networks, 22(3):237–246, 2009.
>
>
>
> **Due to space constraints, we have divided the comment into 2 parts. Please see Part 2.**

---

> > ### Comment · Reviewer_EcSW · 2023-11-20
> > **Thank you for your response**
> >
> > Thank you for your responses.
> > The responses seem to be satisfactory; I have one question here.
> > I meant by "solutions" to the solutions to ODE; and your arguments on Lipschitz continuity etc. make sense.  However, my point was "given that the CT system needs more conditions on the dynamical systems to ensure the existence of solutions to ODE throughout iterations, doesn't it hinder the performance benefits of CT system?".  So as long as there are convincing arguments saying the benefits outweigh the aforementioned restrictions, it is fine I think.
> > (I know the algo does not solve ODE directly but approximately with discrete data, but then it does not guarantee the convergence of the whole algorithm.  Is there any edge case where it diverges because of this fact?)

---

> ### Author Response · Authors · 2023-11-18
> **Response to Reviewer EcSW Part2**
>
> **[Q4: For stochastic systems, CT requires more conditions for certain analyses]**
>
> **Response:**
>
> This is a very important question. In fact, this is the area we're currently delving into. Let me share some analysis:
>
> The analysis of CT stochastic systems is more complex due to the need to consider stochastic differential equations (SDEs) instead of ordinary differential equations (ODEs). Building on Itô's lemma, the optimal control problems for linear stochastic systems can be effectively addressed using policy iteration (PI) as demonstrated in [5]. This PI process involves integral calculations in each iteration, similar to deterministic systems. However, a key distinction lies in the approximation of these integrals for SDEs, which requires numerical methods specifically designed for SDEs, like the Euler–Maruyama method.
>
> Expanding our paper's analysis to include CT stochastic systems, particularly the algorithm in [5], would involve assessing the computational error associated with methods like Euler–Maruyama. However, this is out of the scope of this paper and represents an intriguing avenue for further research.
>
> **[Q5: If we know the utility lives in certain RKHS, can we say anything about the value function which may validate the assumptions]**
>
> **Response:**
>
> This is an important question. The value function in a control system is determined not only by the utility function but also by the underlying dynamical system. This relationship can be verified when we combine the continuous-time (CT) Bellman equation (1a) with the system dynamics (4), leading to the equation:
> $$
> \nabla_x{V}(x)(f(x)+g(x)u) = -l(x(t), u(x(t)).
> $$
> However, the precise formulation of the value function in a general nonlinear system is more complex and challenging to analyze. As of now, we cannot provide a definitive answer.
>
> But, we can restrict our analysis to linear quadratic systems with system dynamics $\dot{x} = Ax + Bu$ and utility function $l(x,u)=x^{\top}Qx+u^{\top}Ru$. In this case, if $[A,B]$ is controllable, the value function is known to be quadratic [1], represented as $V(x)=x^{\top}Px$, where $P\succeq0$. Hence, the value function is continuously differentiable, satisfying the conditions outlined in Assumption 1.
>
> Besides, the conditions for the basis function in Assumption 1 can also be verified. The key idea is to choose the basis function $\phi(x)$ as the non-redundant elements of $x\otimes x$, with $\otimes$ denoting the Kronecker product. Therefore, in this framework, $V(x)$ can be represented as $V(x) = \omega^{\top}\phi(x)$, with $\omega,\phi \in \mathbb{R}^{\frac{n_x(n_x+1)}{2}}$ and $\omega$ represents the non-redundant components of $\mathrm{vec}(P)$. For example, in a two-dimensional system, $\phi(x) := \begin{bmatrix}x_1^2 & x_1x_2 & x_2^2\end{bmatrix}^{\top}$. In such a configuration, $\phi(x)$ fulfils Assumption 1, which encompasses continuity, differentiability, and linear independence.
>
> We also appreciate your pointing out the typo regarding the integral value interval, which has been corrected in the updated version of our paper.
>
> **[Q6: In Appendix H; no approach to find a suitable T? How rare the independence property fails for a random T?]**
>
> In Appendix H, we highlight that a suitable integration interval $T$ is essential to ensure that the matrix $\Theta^{(i)}$ achieves full column rank, which is necessary for the least squares method to produce a unique solution. The full rank of  $\Theta^{(i)}$ is primarily determined by two factors:
>
> 1. **Number of Data Points $m$**: It's advisable to have the number of rows in $\Theta^{(i)}$ be at least twice the dimension of the basis function. Practically, this means ensuring that $m$, the number of data points, is greater than twice $n_{\phi}$. This approach aligns with the guidance provided in [1] (Remark 2.3.2).
> 2. **Integration Interval $T$**: In our experiments, we found that achieving full rank for $\Theta^{(i)}$ in nonlinear systems is generally straightforward. We did not come across any instances where this condition was not met. For linear systems, an integration interval of $T>0.1$ is typically sufficient to satisfy the full rank condition.
>
> **[Q7: For Appendix G, not only for the figure for the utility itself, a plot for the integral of the utility and the worst case error would be informative too.]**
>
> Thank you for your valuable suggestion! We have included the requested figures in Appendix G.
>
> **References**
>
> [5] Tao Bian, Yu Jiang, and Zhong-Ping Jiang. Adaptive dynamic programming for stochastic systems with state and control dependent noise. IEEE Transactions on Automatic Control, 61(12):4170–4175, 2016. doi: 10.1109/TAC.2016.2550518
>
> In closing, we hope that our responses to the comments have adequately addressed your questions and concerns. We deeply appreciate the opportunity to engage in this insightful dialogue, and we are open to any further suggestions or clarifications that may enhance the quality and understanding of our work.

---

> > ### Comment · Reviewer_EcSW · 2023-11-20
> > **Thank you for your response part2**
> >
> > For stochastic systems, if you need to rely on ito lemma, it might not well represent certain behaviors (as the stochasticity of ito diffusion is in restricted form).  Again, it would be great if you could just put a convincing argument in the paper that CT formulation is still more beneficial in certain applications.
> >
> > Overall, thank you for your responses and they are mostly satisfactory to me.

---

> ### Author Response · Authors · 2023-11-20
> **Further Response to Reviewer EcSW**
>
> Thanks for quick response and your insightful suggestions.
>
> **[Q1: Given that the CT system needs more conditions on the dynamical systems to ensure the existence of solutions to ODE throughout iterations, doesn't it hinder the performance benefits of CT system?. So as long as there are convincing arguments saying the benefits outweigh the aforementioned restrictions, it is fine I think.]**
>
> **Response:**
>
> We completely agree with your point. Indeed, solving ODEs in CTRL does require more rigorous conditions, but this rigour translates into greater precision.
>
> When a system is inherently continuous-time, modelling it with CT transition models offers greater precision than DT transition models. For instance, in [1], figure 1 demonstrates the benefits of using CT transition models. It shows a comparison between the true solution of the CartPole system and its approximations in both discrete and continuous-time trajectories. The continuous-time solution closely aligns with the true solution, whereas the discrete trajectory significantly diverges. Similarly, Figure 2-b of [1] presents more evidence supporting the precision of CT models for the Acrobat task. Another crucial scenario to consider involves irregularly sampled data. In such cases, constructing a discrete-time transition model often fails to accurately capture the system's true dynamics, as illustrated in Figure 4 of [1].
>
> These examples suggest that, for many continuous systems, modelling them as discrete transition models is not a good choice. Thus, despite the additional conditions required, the benefits of considering CT transition models in accurately representing CT systems are substantial.
>
> **[Q2: I know the algo does not solve ODE directly but approximately with discrete data, but then it does not guarantee the convergence of the whole algorithm. Is there any edge case where it diverges because of this fact?]**
>
> **Response:**
>
> We completely agree with your perspective.
>
> Intuitively, approximating with discrete data can indeed introduce computational errors. When the data is sparse, these computational errors can be significant. Let's consider an extreme case: if the error is so large that it fails to reflect the value function, then the algorithm would undoubtedly not work effectively.
>
> By now, it is challenging to provide a quantitative analysis specifying the exact conditions under which the algorithm would diverge. However, one explanatory point is that, if the computational error is sufficiently large, it would correspond to a substantial value of  $\bar{\epsilon}$  in Theorem 2. This, in turn, makes it harder to satisfy the convergence conditions (iv) and (v) for Newton's method to converge in Theorem 2.
>
> **[Q3: For stochastic systems, if you need to rely on the ito lemma, it might not well represent certain behaviours (as the stochasticity of ito diffusion is in restricted form. Again, it would be great if you could just put a convincing argument in the paper that CT formulation is still more beneficial in certain applications.]**
>
> **Response:**
>
> We fully agree that Ito diffusion has limitations by now. So, in our previous response, we only provided an example of an SDE that might fit within our analytical framework. Performing CTRL for general CT stochastic systems requires further collaborative efforts from the entire research community.
>
> As for the assertion that CT formulations are more beneficial in certain applications, it's indeed challenging to find direct comparative performance examples between DTRL and CTRL, as such comparisons are rare in academic literature. However, we did find a compelling example in the financial domain in the article [2]. Many models in finance inherently use SDEs. In [2], a comparison is made between a CTRL application (the EMV algorithm presented in the paper) and a DTRL application (DDPG). The EMV algorithm demonstrated superior performance compared to DDPG. This instance supports that CTRL can offer significant advantages in scenarios where the underlying models are SDEs.
>
> We acknowledge the importance of including these arguments in our paper and will summarise and incorporate them into the updated version.
>
> **References**
>
> [1] Cagatay Yildiz, Markus Heinonen, and Harri L ̈ahdesm ̈aki. Continuous-time model-based reinforcement learning. In International Conference on Machine Learning, pp. 12009–12018. PMLR, 2021.6
>
> [2] Haoran Wang and Xun Yu Zhou. Continuous-time mean–variance portfolio selection: A reinforcement learning framework. Mathematical Finance, 30(4):1273–1308, 2020.
>
> **Thank you once again for your active participation in this discussion and for your prompt response. Your insights are immensely valuable to us.**

---

> > ### Comment · Reviewer_EcSW · 2023-11-21
> > **Thank you**
> >
> > Thank you for your response again.
> >
> > Those are all satisfactory to me; I keep my score but I recommend acceptance of this paper.

---

> > > ### Author Response · Authors · 2023-11-21
> > > **Thank you for engaging in this discussion.**
> > >
> > > It's been a pleasure discussing with you. We're pleased to have resolved your concerns. Your insightful review has significantly contributed to the refinement of our work, and we're grateful for your guidance in enhancing the strength of our paper.

---

### Official Review · Reviewer_jx6Y · 2023-10-30

**Soundness:** 3 good
**Presentation:** 2 fair
**Contribution:** 2 fair
**Rating:** 6
**Confidence:** 2

**Summary:**

The paper addresses the impact of computational methods on control performance in Integral Reinforcement Learning. The authors focus on the policy evaluation stage of IntRL, where the integral of the utility function needs to be computed using quadrature rules. They demonstrate that computational errors introduced during PEV can significantly influence the convergence behavior of policy iteration and the performance of the learned controller.

The authors show that computational errors in PEV manifest as an extra error term in each iteration of Newton's method. They provide a theoretical analysis, proving that the upper bound of this error term is proportional to the computational error. The paper further explores the case where the utility function resides in a reproducing kernel Hilbert space (RKHS), presenting local convergence rates for IntRL using both the trapezoidal rule and Bayesian quadrature with a Matern kernel.

**Strengths:**

- The authors demonstrated how computational errors in the PEV stage of IntRL affect the convergence behavior of policy iteration and the performance of the learned controller, which is previously unexplored.
- They also provided a solid theoretical analysis of the impact of computational errors, providing bounds and convergence rates that relate the computational method to control performance.
- Validation of the theoretical findings is also offered through simulations on canonical control tasks, showing the practical implications of the choice of computational method in IntRL.

The paper sheds light on the impact of computational methods on control performance in IntRL, providing both theoretical insights and practical guidelines for improving controller learning in continuous-time reinforcement learning scenarios.

**Weaknesses:**

- The paper provides theoretical claims about the impact of computational methods on control performance in IntRL. However, the experimental validation seems to be limited in scope. The authors only consider canonical control tasks to validate their findings. The authors could consider a broader set of experiments, including more complex and real-world scenarios, to showcase the practical implications of their findings.

**Questions:**

Could you provide more details on the choice of the canonical control tasks used for experimental validation? Were any real-world scenarios or more complex tasks considered?

---

> ### Author Response · Authors · 2023-11-18
> **Response to Reviewer jx6Y**
>
> Thank you for acknowledging the insights in our paper and for your valuable feedback！
>
> **[Q1: The authors could consider a broader set of experiments, including more complex and real-world scenarios, to showcase the practical implications of their findings. Were any real-world scenarios or more complex tasks considered?]**
>
> **Response:**
>
> Thank you for this important question. To apply to high-dimensional and real-world scenarios will definitely make a huge impact. In fact, we did try some complicated tasks such as drones, autonomous vehicles, etc. Unfortunately, we can't train the algorithm to converge; thus, it wasn't shown in this paper.
>
> We analyze the failure and recognize that this is mainly due to the inherent limitation of the integral RL (IntRL) algorithm. IntRL is a canonical control task in CTRL, just like the Q learning in DTRL. However, IntRL is still underdeveloped for high-dimensional systems compared to Q learning. As shown in [1], [2], and [3], IntRL is typically applied to systems with no more than 4 dimensions. Additionally, we wish to emphasize that our paper primarily focuses on analyzing the phenomenon of "computation impacts control" rather than on applying IntRL in complex systems.
>
> However, since we recognized the importance of high-dimensional applications, we have considered the challenges associated with this. Specifically, the difficulties in achieving convergence in high-dimensional systems can be attributed to several factors:
>
> 1. **Limited Approximation Capability**: IntRL uses a linear combination of basis functions to approximate the value function, which is less powerful than the neural network-based approaches prevalent in DTRL.
> 2. **Lack of Effective Exploration Mechanisms**: IntRL struggles with poor data distribution due to inadequate exploration strategies. Common DTRL methods improve exploration by adding noise during data collection, but in continuous systems, this leads to complex stochastic differential equations (SDEs) that are challenging to manage.
> 3. **Absence of Advanced Training Techniques**: Techniques like replay buffers, parallel exploration, delayed policy updates, etc., which enhance sample efficiency and training in DTRL, are lacking in IntRL.
> 4. **No Discount Factor**: The absence of a discount factor in IntRL makes it difficult to ensure that policy iteration acts as a contraction mapping, which can affect the stability of the algorithm's training process.
> 5. **Non-existence of Q function**: In CT systems, the concept of a Q function is not directly applicable. The absence of Q function impedes the direct translation of many DTRL methodologies and insights to IntRL.
>
> Considering these challenges, we will investigate and develop CTRL algorithms for complex and high-dimensional scenarios in future research.
>
> Since this is an important question, we add Appendix K in our paper to discuss the limitations of IntRL.
>
> **[Q2: Could you provide more details on the choice of the canonical control tasks used for experimental validation?]**
>
> **Response:**
>
> Thank you for your insightful question. Our selection of canonical control tasks was intentionally based on seminal works within the field of IntRL. Example 1 (linear system) in our paper is adapted from Example 1 in Chapter 2 of [4], and Example 2 (nonlinear system) is based on Example 1 in [1]. These specific examples were chosen for their significance and representational value in the established IntRL literature.
>
> Our research is primarily centered on introducing and demonstrating the "computation impacts control" phenomenon using integral reinforcement learning (IntRL) algorithms. We acknowledge the challenges in extending IntRL to high-dimensional systems, and this is a limitation of our current approach. We hope that in the future, together with the community, we can further broaden its application to more complex systems.
>
>
>
> **References**
>
> [1] Draguna Vrabie and Frank Lewis. Neural network approach to continuous-time direct adaptive optimal control for partially unknown nonlinear systems. Neural Networks, 22(3):237–246, 2009.
>
> [2] Hamidreza Modares, Frank L Lewis, and Mohammad-Bagher Naghibi-Sistani. Integral reinforcement learning and experience replay for adaptive optimal control of partially-unknown constrained-input continuous-time systems. Automatica, 50(1):193–202, 2014.
>
> [3] Brent A Wallace and Jennie Si. Continuous-time reinforcement learning control: A review of theoretical results, insights on performance, and needs for new designs. IEEE Transactions on Neural Networks and Learning Systems, 2023.
>
> [4] Yu Jiang and Zhong-Ping Jiang. Robust adaptive dynamic programming. John Wiley & Sons, 2017.
>
> In closing, we hope that our responses to the comments have adequately addressed your questions and concerns. We are open to any further suggestions or clarifications. Thank you for your valuable input, which has been instrumental in enriching our research.

---

> > ### Comment · Reviewer_jx6Y · 2023-11-21
> > **Thank you for the response**
> >
> > I appreciate the detailed response from the authors.

---

> > > ### Author Response · Authors · 2023-11-22
> > > **Thanks for your reponse!**
> > >
> > > We are glad to hear that you are satisfied with our response. If you have any more questions, feel free to ask.

---

### Official Review · Reviewer_3VMw · 2023-11-05

**Soundness:** 3 good
**Presentation:** 3 good
**Contribution:** 3 good
**Rating:** 6
**Confidence:** 2

**Summary:**

This paper studies the continuous time RL and provides a detailed convergence rate discussion on the impact of policy iteration's computation errors when approximating the integration in the policy evaluation step. By showing that PI can be viewed as Newton updates and PI with computation errors can be viewed as Newton updates with errors, this paper established local convergence rates and uses simulation to demonstrate the tightness of the order of the convergence rate.

**Strengths:**

The paper is very well written. Even though I am not an expert in this area, the detailed motivation and the clear illustration diagrams help me understand the importance of this problem and the key ideas behind the proofs. The connection between Newton updates and PI, and using approximate Newton to analyze PI with computation errors are also fascinating. Further, the proofs are quite involved too. Lastly, the numerical results demonstrate the tightness of the order of the convergence rate with respect to N.

**Weaknesses:**

See below.

**Questions:**

Q1: In Theorem 3, the computation error is treated as a constant value. How does this constant decay with the number of samples?

Q2: Corollary 1 assumes that $i \to +\infty$. By using the decay rate of the computation error in Theorem 1, can the authors comment on a more realistic convergence rate based on different number of iterations $i$? Further, can the theoretical results provide some guidelines on how to choose the number of iterations to terminate at?

Q3: How does this result compare with LSPI, which is also based on a linear combination of basis functions?

---

> ### Author Response · Authors · 2023-11-18
> **Response to Reviewer 3VMw**
>
> Thank you for acknowledging the insights in our paper and for your valuable feedback！
>
> **[Q1: In Theorem 3, the computation error is treated as a constant value. How does this constant decay with the number of samples?]**
>
> **Response:**
>
> The constant value $\bar{\epsilon}$, representing the upper bound of the extra error term $E_i$ in  Newton's method, is directly linked to the computational error in the integrals of matrix $\Xi^{(i)}$.
>
> As we detailed in Section 3.2 of our paper, the computational error of the integral is bound by the product of the integrand’s norm in the reproducing kernel Hilbert space (RKHS) and the worst-case error, as formulated in equation (13). Importantly, the worst-case error diminishes as the number of samples $N$ increases.
>
> When Bayesian quadrature (BQ) is employed as the quadrature rule, the worst-case error for Wiener and Matérn kernels is reduced at rates of $𝑂(𝑁^{-2})$ and $𝑂(𝑁^{-b})$ respectively, where $b$ is the smoothness parameter of the Matérn kernel.
>
> Thus, the constant value $\bar{\epsilon}$ decays with the number of samples at a rate of $O(N^{-2})$ for Wiener kernel and $O(N^{-b})$ for Matérn kernel, depending on the kernel's smoothness parameter $b$.
>
> **[Q2: Can the authors comment on a more realistic convergence rate based on different number of iterations $i$? Further, can the theoretical results provide some guidelines on how to choose the number of iterations to terminate at?]**
>
> **Response:**
>
> Yes, of course! According to Theorem 3, the convergence rate considering the number of iterations can be described as $O(2^{-i})+O(N^{-b})$, where $i$ represents the algorithm's iteration steps, and $N$ is the sample size. The term $O(2^{-i})$ indicates the error due to insufficient iterations for convergence. Since this error decays exponentially with increasing $i$, it rapidly approaches zero.
>
> In simulations, especially for low-dimensional systems, we generally find that $i\geq10$ ensures convergence.
>
> **[Q3: How does this result compare with LSPI, which is also based on a linear combination of basis functions?]**
>
> **Response:**
>
> Thank you for this intriguing question. Indeed, when referring to least-squares policy iteration (LSPI) [1], there are similarities between LSPI and Integral RL (IntRL) in terms of approximating the value function using linear basis functions. From this perspective, IntRL can be viewed as a continuous-time (CT) counterpart of LSPI.
>
> LSPI employs the least squares method to solve for the Q function, while IntRL focuses on the V function.  However, the concept of a Q function can not be applicable to CT systems [2]. This is because in the context of CT systems, if we were to define the Q function as it is in discrete-time systems, then the control input can be instantaneously switched to an optimal control without impacting the overall cost.
>
> Therefore, LSPI is not directly applicable to CT systems, and the IntRL algorithm itself can be seen as the CT version of LSPI.
>
> **References**
>
> [1] Michail G Lagoudakis and Ronald Parr. Least-squares policy iteration. The Journal of Machine Learning Research, 4:1107–1149, 2003
>
> [2] Frank L Lewis and Draguna Vrabie. Reinforcement learning and adaptive dynamic programming for feedback control. IEEE circuits and systems magazine, 9(3):32–50, 2009
>
>
>
> In closing, we hope that our responses to the comments have adequately addressed your questions and concerns. We deeply appreciate the opportunity to engage in this insightful dialogue, and we are open to any further suggestions or clarifications that may enhance the quality and understanding of our work. Thank you for your valuable input, which has been instrumental in enriching our research.

---

### Official Review · Reviewer_8W8E · 2023-11-08

**Soundness:** 3 good
**Presentation:** 3 good
**Contribution:** 2 fair
**Rating:** 8
**Confidence:** 3

**Summary:**

This paper studies the impact of computation methods (quadrature rule for solving integrals) when applying reinforcement learning in continuous control tasks. Building upon the connections between HJB equation and Newton's method, the authors show that the computation error is an extra error term in each iteration of Newton's method. With the bounded error assumption, they provide a convergence results for Newton's methods with an extra error term (Theorem 1). Furthermore, the computation error bounds are also discussed by minimizing the worst case error under different quadrature rules and kernels. Finally, an end to end convergence result is provided (Theorem 3, Corollary 1).

**Strengths:**

This paper is well written and easy to follow. The problem of studying the impact of computational errors on continuous control in integral RL is well motivated and interesting. As the authors claim, this problem is widespread but understudied.

**Weaknesses:**

Although the problem studied in this work is interesting. I feel like the paper is mainly a combination of existing results (convergence of Newton's method with an extra error term and error bound on the computation step). What are the novel techniques appled in the analysis such that this work is not simply A+B?

**Questions:**

1. Are the affine nonlinear systems necessary? Can you consider a more general class of systems?
2. The assumptions of Theorem 2 need more explanations. To guarantee those assumptions, what properties do you need for the systems functions $f$, $g$, and cost function $J$?
3. The experimental examples are basically toy examples (3d linear system and 2d nonlinear system). These are far from high-dimensional real-time applications. How the computation error and convergence will behave in higher dimensional cases need to be examined.

---

> ### Author Response · Authors · 2023-11-18
> **Response to Reviewer 8W8E Part 1**
>
> Thank you for your valuable comments！
>
> **[Q1: I feel like the paper is mainly a combination of existing results (convergence of Newton's method with an extra error term and error bound on the computation step). What are the novel technique applied in the analysis such that this work is not simply A+B?]**
>
> **Response:**
>
> Thank you for this question, which provides an opportunity to clarify the unique contributions of our paper.
>
> The core insight of our paper is the discovery and analysis of the "computation impacts control" phenomenon within the integral reinforcement learning (IntRL) algorithm.
>
> Our work is not a mere combination of existing results. We began by finding the "computation impacts control" phenomenon in simulations.  To understand this phenomenon, we hypothesized that different computational methods could lead to varying accuracies in computing the integral in policy evaluation, thereby impacting the convergence of policy iteration and, ultimately, control performance. Thus, we try to find some theoretical analysis to support our hypothesis.
>
> However, we noted that integral computation during policy evaluation has been largely overlooked in existing research. For example, widely-used open-source codes, like those in reference [3], default to the Runge-Kutta 45 method for this computation. This method is not suitable, or essentially incorrect, in scenarios where system internal dynamics are unknown, as the analytical form of the integrand is unavailable.
>
> This led us to explore optimal quadrature rules for integral computation using state sample data and quantify the computational error.  Our research found that Bayesian quadrature (BQ) is an optimal approach. Additionally, BQ's computational error bounds can be effectively characterized by the variance obtained through BQ. Thus, we introduce BQ into IntRL for integral computation in policy evaluation, providing a tangible upper bound for computational error.
>
> We then analyzed the convergence of IntRL with this bounded computational error. However, this presented significant challenges. The convergence analysis of IntRL is more complex than in discrete-time RL (DTRL) because, in DTRL, a discount factor $\gamma$ simplifies the analysis. This allows the Bellman operator to be treated as a contraction mapping, bounding computational errors during iterations, as discussed in Section 6.2 of [1]. In continuous-time RL (CTRL), the absence of $\gamma$ introduces challenges in analyzing convergence, particularly when considering errors like learning errors or computational errors. This is why most convergence analyses in CTRL, including those in Theorem 4 of [2] and Theorem 3.2.3 of [3], assume negligible or zero learning errors and computational errors.
>
> To address this, we treated policy iteration as an iteration of Newton's method, enabling a novel analysis of local convergence that accounts for an extra error term. Analyzing the convergence of Newton's method with a bounded error term is also technically challenging. Existing works consider different forms of errors, such as floating or rounding errors [4][5], which are not directly applicable in CTRL scenarios. Works closest to our setting, like [6], rely on Lipschitz bounded assumptions for convergence, which are not valid in CTRL. Hence, our approach required rigorous derivation and proof, underscoring the technical complexity and originality of our work.
>
> **References**
>
> [1] Dimitri Bertsekas and John N Tsitsiklis. Neuro-dynamic programming. Athena Scientific, 1996. 6
>
> [2] Draguna Vrabie and Frank Lewis. Neural network approach to continuous-time direct adaptive optimal control for partially unknown nonlinear systems. Neural Networks, 22(3):237–246, 2009.
>
> [3] Yu Jiang and Zhong-Ping Jiang. Robust adaptive dynamic programming. John Wiley & Sons, 2017.
>
> [4] Francoise Tisseur. Newton’s method in floating point arithmetic and iterative refinement of generalized eigenvalue problems. SIAM Journal on Matrix Analysis and Applications, 22(4):1038–1057, 2001
>
> [5] Tjalling J Ypma. The effect of rounding errors on newton-like methods. IMA Journal of Numerical Analysis, 3(1):109–118, 1983.
>
> [6] Minoru Urabe. Convergence of numerical iteration in solution of equations. Journal of Science of the Hiroshima University, Series A (Mathematics, Physics, Chemistry), 19(3):479–489, 1956.
>
>
>
> **Due to space constraints, we have divided the comment into 3 parts. Please see Part 2.**

---

> ### Author Response · Authors · 2023-11-18
> **Response to Reviewer 8W8E Part 2**
>
> **[Q2: Are the affine nonlinear systems necessary? Can you consider a more general class of systems?]**
>
> **Response:**
>
> The original research on IntRL indeed focused primarily on affine nonlinear systems. In fact, most continuous-time reinforcement learning (CTRL) papers, such as [2], [3], and [7], concentrate on control affine systems with quadratic costs. This focus is largely due to the fact that, in such scenarios, policy improvement method (PIM) does not require knowledge of the system's internal dynamics and provides an explicit solution (as detailed in equation (8b) of our paper).
>
> However, when the internal dynamics of a system is known, IntRL algorithms can indeed be extended to a broader class of systems and cost functions. Consider a general nonlinear system with dynamics $\dot{x}=f(x,u)$ with a cost function $l(x,u)$.   In this case, the policy iteration (PI) process in CTRL involves:
>
> **Policy evaluation (PEV):** Evaluate the value of the control policy using
> $$
> V^{(i)}\left(x(t)\right) = \int_{t}^{t+\Delta{T}} l(x(s), u^{(i)}(x(s)) \\,\mathrm{d}s + V^{(i)}\left(x(t+\Delta{T})\right).
> $$
> **Policy improvement (PIM):** Determine an improved policy
> $$
> u^{(i+1)}(x) = \underset{u}{\arg\min} \\left\\{l(x,u) + (\nabla_x {V^{(i)})^{\top}f(x,u)}\\right\\}.
> $$
> The corresponding Hamilton-Jacobi-Bellman (HJB) equation is
> $$
> V^{\star}(x(t)) = \min_{u(t:t+T)} \\left\\{ \int_t^{t+T}l(x(s),u(s)) \\,\\mathrm{d}s+ V^{\star}(x(t+T))\\right\\}.
> $$
>
> Technically, the convergence analysis in our paper is also limited to control affine systems with quadratic costs due to our approach of treating the HJB equation as an iteration of Newton's method. For such systems, the HJB equation simplifies to a function of $V$, i.e., $G(V)=0$  (as shown in equation (7)). However, for general nonlinear systems, their corresponding HJB equations are more complex, making the transformation of PI iteration into Newton's method iteration unfeasible.
>
> In summary, to better elucidate our theory, we restrict our analysis to control affine systems. The convergence analysis of general nonlinear systems involves more intricate mathematical tools and is a subject for future exploration.
>
>
>
> **[Q3: The assumptions of Theorem 2 need more explanations. To guarantee those assumptions, what properties do you need for the systems functions $f$, $g$, and cost function $J$?]**
>
> **Response:**
>
> Thank you for highlighting the need for more detailed explanations of Theorem 2's assumptions.
>
> Regarding the first part of the question, the central assumption of Theorem 2 is the selection of an appropriate initial control policy $u^{(0)}$, which ensures that the mapping  $G(V)$ meets specific criteria in the vicinity of the value function $V^{(0)}$. This vicinity corresponds to the range covered by the first iteration of Newton's method. We aim for the mapping $G(V)$ to maintain second-order continuity (Assumption i), have invertible derivatives (Assumption iii), and ensure that these derivatives are sufficiently large (Assumptions ii and iv).  Furthermore, when the computational error $\bar{\epsilon}$ is small enough, Assumptions (iv) and (v) are more easily satisfied.
>
> Regarding the second part of the question, we have to admit that we don't have a definite answer to show the properties yet. Now we try to explain it qualitatively.
>
> Determining the quantitative relationship between the system's properties and the assumptions in Theorem 2 is a challenging question. Current convergence conclusions predominantly build on the findings in [8], which state that convergence is achievable if $f$ and $g$ are Lipschitz continuous and the initial control policy is admissible. However, the framework in [8] does not account for computational errors, this is precisely why we did not directly use their conclusions in our analysis. A deeper exploration into this, potentially building upon [8]'s analysis to outline necessary system assumptions, would be a significant challenge at now. We think we require more advanced mathematical tools to analyze this question. We admit that establishing a quantitative relationship would significantly improve our understanding of the relationship between policy iteration, dynamic systems, and Newton's method. So we plan to tackle this complex task in future work.
>
> However, it's important to note that even if the conditions of Theorem 2 cannot be directly translated into checkable system characteristics, our proof remains rigorous and the conclusions valid.
>
> **References**
>
> [7] Hamidreza Modares, Frank L Lewis, and Mohammad-Bagher Naghibi-Sistani. Integral reinforcement learning and experience replay for adaptive optimal control of partially-unknown constrained-input continuous-time systems. Automatica, 50(1):193–202, 2014.
>
> [8] Randal W Beard, George N Saridis, and John T Wen. Galerkin approximations of the generalized hamilton-jacobi-bellman equation. Automatica, 33(12):2159–2177, 1997.
>
> **Please see Part 3.**

---

> ### Author Response · Authors · 2023-11-18
> **Response to Reviewer 8W8E Part 3**
>
> **[Q4: The experimental examples are basically toy examples (3d linear system and 2d nonlinear system). These are far from high-dimensional real-time applications. How the computation error and convergence will behave in higher dimensional cases need to be examined.]**
>
> **Response:**
>
> This is the same question as commented by reviewer jx6Y. So we just copy the answer.
>
> To apply to high-dimensional and real-world scenarios will definitely make a huge impact. In fact, we did try some complicated tasks such as drones, autonomous vehicles, etc. Unfortunately, we can't train the algorithm to converge; thus, it wasn't shown in this paper.
>
>
>
> We analyze the failure and recognize that this is mainly due to the inherent limitation of the integral RL (IntRL) algorithm. IntRL is a canonical control task in CTRL, just like the Q learning in DTRL. However, IntRL is still underdeveloped for high-dimensional systems compared to Q learning. As shown in [1], [9], and [10], IntRL is typically applied to systems with no more than 4 dimensions. Additionally, we wish to emphasize that our paper primarily focuses on analyzing the phenomenon of "computation impacts control" rather than on applying IntRL in complex systems.
>
> However, since we recognized the importance of high-dimensional applications, we have considered the challenges associated with this. Specifically, the difficulties in achieving convergence in high-dimensional systems can be attributed to several factors:
>
> 1. **Limited Approximation Capability**: IntRL uses a linear combination of basis functions to approximate the value function, which is less powerful than the neural network-based approaches prevalent in DTRL.
> 2. **Lack of Effective Exploration Mechanisms**: IntRL struggles with poor data distribution due to inadequate exploration strategies. Common DTRL methods improve exploration by adding noise during data collection, but in continuous systems, this leads to complex stochastic differential equations (SDEs) that are challenging to manage.
> 3. **Absence of Advanced Training Techniques**: Techniques like replay buffers, parallel exploration, delayed policy updates, etc., which enhance sample efficiency and training in DTRL, are lacking in IntRL.
> 4. **No Discount Factor**: The absence of a discount factor in IntRL makes it difficult to ensure that policy iteration acts as a contraction mapping, which can affect the stability of the algorithm's training process.
> 5. **Non-existence of Q function**: In CT systems, the concept of a Q function is not directly applicable. The absence of Q function impedes the direct translation of many DTRL methodologies and insights to IntRL.
>
> Considering these challenges, we will investigate and develop CTRL algorithms for complex and high-dimensional scenarios in future research.
>
> Since this is an important question, we add Appendix K to our paper to discuss the limitations of IntRL.
>
> **References**
>
> [9] Hamidreza Modares, Frank L Lewis, and Mohammad-Bagher Naghibi-Sistani. Integral reinforcement learning and experience replay for adaptive optimal control of partially-unknown constrained-input continuous-time systems. Automatica, 50(1):193–202, 2014.
>
> [10] Brent A Wallace and Jennie Si. Continuous-time reinforcement learning control: A review of theoretical results, insights on performance, and needs for new designs. IEEE Transactions on Neural Networks and Learning Systems, 2023.
>
> In closing, we hope that our responses to the comments have addressed your questions and concerns. We deeply appreciate the opportunity to engage in this insightful dialogue, and we are open to any further suggestions or clarifications that may enhance the quality and understanding of our work. Thank you for your valuable input, which has been instrumental in enriching our research.

---

> ### Comment · Reviewer_8W8E · 2023-11-20
> **Thank you for the detailed response**
>
> My concerns have been resolved by the authors' response, especially on the novelty of this work. Now I am convinced that this work is not simply an A+B and could be appreciated by the community. In light of this, I am increasing my score to 8.

---

> ### Author Response · Authors · 2023-11-21
> **Thanks for your valuable comments!**
>
> We're delighted to have addressed your concerns. Your insightful review has been instrumental in refining our work. Thanks for your guidance in making our paper stronger!

---

### Author Response · Authors · 2023-11-21
**Revision Summary**

We sincerely appreciate the detailed and insightful feedback provided by the reviewers. Their valuable comments have been crucial in enhancing the quality of our paper. In response to their recommendations, we have implemented the following key modifications:

* Add appendix K to discuss the limitations of the IntRL algorithm
* Add appendix L to more thoroughly discuss the motivation of the CTRL algorithms

Please note that all major revisions are highlighted in blue for easy identification.

---

> ### Author Response · Authors · 2023-11-21
> **Thanks to Reviewers and ACs**
>
> Dear reviewers and ACs,
>
> We extend our heartfelt thanks for the time and effort you devoted to reviewing our manuscript. Your perceptive comments and valuable suggestions have significantly contributed to the enhancement of our paper's quality and clarity. We have attentively integrated your feedback and made efforts to address all raised concerns.
>
> We hope that our revisions and responses clearly convey our work's intent and address your points. Should there be any remaining issues or queries, please feel free to raise them. We are more than willing to provide further clarifications.
>
> We deeply value your support and guidance, and are thankful for your role in furthering research in our field.
>
> Warm regards,
>
> Authors

---

### Meta-Review · Area_Chair_ZNT9 · 2023-12-06

**Metareview:**

The paper investigates the impact of computation methods (quadrature rule for solving integrals) when using reinforcement learning in continuous control tasks. It provides a new angle of studying RL in continuous control, is very well-written, and contains solid and interesting theoretical results (corroborated with convincing experimental results). It reaches a consensus that this is a good paper that is worth acceptance.

**Justification For Why Not Higher Score:**

Although the paper has several outstanding features, its focus on continuous-time setting may not have as general applicability (in terms of techniques being used, and some insights in the results) to the more relevant setting of discrete-time RL in the ICLR/ML community. Also, the experimental section could have been stronger if it had not been restricted to the canonical control examples only.

**Justification For Why Not Lower Score:**

It is definitely a good paper above the acceptance bar, and is a novel one that can attract more research attention in this sub-area. The authors were also very dedicated ito responding to the comments (tho most of them are positive ones).

---

### Decision · Program_Chairs · 2024-01-16

Accept (spotlight)